# Dominance vs epistasis: the biophysical origins and plasticity of genetic interactions within and between alleles

Xuan Xie[1], Xia Sun[1,8], Yuheng Wang [1,8], Ben Lehner [2,3,4,5] ✉ & Xianghua Li[1,5,6,7] ✉

An important challenge in genetics, evolution and biotechnology is to understand and predict how mutations combine to alter phenotypes, including molecular activities, fitness and disease. In diploids, mutations in a gene can combine on the same chromosome or on different chromosomes as a "heteroallelic combination". However, a direct comparison of the extent, sign, and stability of the genetic interactions between variants within and between alleles is lacking. Here we use thermodynamic models of protein folding and ligand-binding to show that interactions between mutations within and between alleles are expected in even very simple biophysical systems. Protein folding alone generates within-allele interactions and a single molecular interaction is sufficient to cause between-allele interactions and dominance. These interactions change differently, quantitatively and qualitatively as a system becomes more complex. Altering the concentration of a ligand can, for example, switch alleles from dominant to recessive. Our results show that intra-molecular epistasis and dominance should be widely expected in even the simplest biological systems but also reinforce the view that they are plastic system properties and so a formidable challenge to predict. Accurate prediction of both intra-molecular epistasis and dominance will require either detailed mechanistic understanding and experimental parameterization or brute-force measurement and learning.

A fundamental goal in biology is to understand and predict how mutations combine to alter phenotypes. This is important in biotechnology – for example when engineering new enzymatic activities and protein properties – and also in animal and plant breeding, clinical genetics and evolutionary biology.

Although mutations are normally assumed to have independent effects, this often proves not to be the case: additional variants within the same gene as well as in other genes can quantitatively and qualitatively alter the impact of a mutation[1]. Predicting these genetic interactions between variants and so improving genetic prediction beyond the performance that can be achieved using additive models is a central challenge in clinical genetics, evolutionary biology, agriculture and biotechnology.

[1]Zhejiang University - University of Edinburgh Institute, Zhejiang University School of Medicine, Haining 314400, P. R. China. [2]Center for Genomic Regulation (CRG), The Barcelona Institute of Science and Technology, Dr. Aiguader 88, Barcelona 08003, Spain. [3]Universitat Pompeu Fabra (UPF), Barcelona 08003, Spain. [4]ICREA, Pg. Luis Companys 23, Barcelona 08010, Spain. [5]Wellcome Sanger Institute, Wellcome Genome Campus Hinxton, Cambridge CB10 1SA, UK. [6]Deanery of Biomedical Sciences, College of Medicine & Veterinary Medicine, University of Edinburgh, Edinburgh EH8 9XD, UK. [7]Biomedical and Health Translational Centre of Zhejiang Province, Haizhou East Road 718, Haining 314400, P. R. China. [8]Present address: Deanery of Biomedical Sciences, College of Medicine & Veterinary Medicine, University of Edinburgh, Edinburgh EH8 9XD, UK. ✉e-mail: bl11@sanger.ac.uk; xl5@sanger.ac.uk

When two variants occur in the same gene in a diploid species, they can either both occur on the same chromosome or each variant can be on a different chromosome, i.e., with one variant on the paternal and one on the maternal chromosome as different alleles. In any individual, approximately 1 in 10 human genes carry two or more variants compared to a reference genome[2], with >20,000 combinations of variants within the same gene observed >1.5 million times in 59 medically actionable genes in ~50k individuals from the UK Biobank population[3]. It is therefore important for clinical genetics to understand and be able to predict what happens when multiple mutations occur in the same gene, and how the outcome differs depending upon whether the two variants occur on the same chromosome ('within allele') or on different chromosomes ('between alleles').

In quantitative genetics, clinical genetics, functional genomics, animal breeding and evolutionary biology, the phenotypic change when combining mutations is most often assumed to be log-additive ($W_{exp\_log}$) (Eq. 1) or additive ($W_{exp\_add}$) (Eq. 2)[1,4]. For example, when two variants (variant $A$ and variant $B$ for example compared to the wild-type variant WT) are combined within the same copy of a gene, the change in phenotype or fitness ($W$) is often expected to be log-additive (Eq. 1), with deviance from this expectation ($W_{exp}$) referred to as a genetic interaction or epistasis ($E$ or $e$ for log-additive or additive model respectively) (Eqs. 3, 4)[1,5].

$$\log\left(W_{exp\_log}\right) = \log(W_A) + \log(W_B) - \log(W_{WT}) \quad (1)$$

$$W_{exp\_add} = W_A + W_B - W_{WT} \quad (2)$$

$$E_{AB} = \log(W_{AB}) - \log\left(W_{exp}\right) \quad (3)$$

$$e_{AB} = W_{AB} - W_{exp} \quad (4)$$

Similarly, when two variants are combined in different alleles ($\alpha^A$ and $\alpha^B$ for example as two different variants of the same gene), the expected phenotype is normally considered to be the average of the phenotypes when the variants are present in two copies (i.e. homozygotes) (Eq. 5), with any deviance from this additive expectation $W_{exp\_\alpha^A/\alpha^B}$ quantified as the dominance index[6,7] or degree of dominance (Eq. 6)[8,9]. The former uses one allele as the reference and scores above or below 0.5 indicate the reference allele is recessive or dominant, respectively. On the other hand, the degree of dominance does not set a reference allele and 0 indicates no dominance, a positive value indicates that the allele with better function is dominant, and vice versa. Complete dominance or recessivity (where the heterozygote phenotype is the same as either parent) leads to an absolute value of 1.

$$W_{exp\_\alpha^A/\alpha^B} = \frac{W_{\alpha^A/\alpha^A} + W_{\alpha^B/\alpha^B}}{2} \quad (5)$$

$$\text{Degree of dominance} = \frac{W_{\alpha^A/\alpha^B} - W_{exp\_\alpha^A/\alpha^B}}{\left|W_{\alpha^B/\alpha^B} - W_{exp\_\alpha^A/\alpha^B}\right|} \quad (6)$$

Mutations have been found to vary extensively in their dominance, with important implications for breeding, evolution and human clinical genetics[10,11].

Various mechanisms have been proposed to cause epistasis, dominance, or both[1,8,11,12]. These include the non-linear relationships between additive free energies and phenotypes. For example, two-state cooperative protein folding results in a sigmoidal relationship between the folding energy of a protein and the fraction of a protein that is folded[13–16]. Additional non-linear relationships between

genotype and phenotype are introduced by molecular interactions[17,18], cooperativity[19,20], molecular competition[21,22], metabolic flux[7,23] and feedback loops and other dynamics in cellular networks[8]. In addition to such 'global' or 'non-specific' interactions between mutations due to nonlinear genotype-phenotype relationships[1], mutation-specific causes of epistasis and dominance are also observed. These include non-additive changes in free energy when mutating energetically-coupled residues[24] and gain-of-function, change-of-function and dominant negative mutations[25].

Protein (or RNA) folding and binding to ligands constitute the fundamental reactions common to nearly all cellular processes. Thermodynamic models have been used to interpret and predict how mutations combine in large-scale experimental datasets[1,18,21,26–29], but a direct comparison of how these foundational biophysical processes cause interactions within (intra-molecular epistasis) and between alleles of a gene is lacking.

Comparisons of how variants interact within and between alleles of a gene are complicated by the different metrics that are typically used to quantify these types of genetic interactions. While between-allele interactions are typically quantified in a diploid system as dominance by comparing the heterozygous and homozygous phenotypes (Eqs. 5, 6), within-allele interactions are quantified using metrics of epistasis (Eqs. 3, 4). In addition, whereas intra-molecular epistasis is most often quantified using a metric that quantifies the deviance from a log-additive expectation (Eqs. 1, 3), dominance is typically quantified as the deviance from an additive expectation (Eqs. 5, 6). This makes direct comparisons of how variants interact within and between alleles confusing, simply because the metrics used are different. Throughout this manuscript, we, therefore, quantify interactions within and between alleles using identical measures of genetic interaction, and we do so using both additive and log-additive expected outcomes. To allow comparisons to standard definitions of dominance, we also quantify the dominance of alleles using the degree of dominance (Eq. 6).

Our results show that, even in the simplest biophysical systems, mutations are expected to frequently combine with outcomes that differ from the additive or log-additive expectations. Moreover, the expected outcome is typically different when combining variants within versus between alleles. In the simplest possible protein system, where a phenotype is linearly dependent on the concentration of a folded protein, there are no between-allele interactions or dominance, but abundant within-allele interactions (intra-molecular epistasis). Adding a single ligand-binding reaction to the system is sufficient to generate between-allele interactions and dominance which change both quantitatively and qualitatively depending on how much ligand is present. Furthermore, in this simple biophysical system, the epistasis depends on the biophysical effects of mutations whether the folding energy or the binding energy is altered, whereas the between-allele interactions and dominance are the same for mutations altering folding or binding. Introducing a nonlinear dependency of a phenotype on the concentration of a protein or protein-ligand complex transforms the within- and between-allele interactions and often in opposite directions. Our results show that intra-molecular epistasis interactions and dominance are expected in even the simplest one-gene diploid biophysical systems but that they are plastic with their magnitude and sign dependent on the precise molecular details of the system and on the conditions. Moreover, they highlight that both within and between-allele genetic interactions and dominance are difficult to predict quantitatively and qualitatively as they depend on the parameter values in even very simple biophysical systems. Prediction of these interactions and dominance will therefore require either detailed mechanistic understanding and quantification of relevant cellular parameters or large-scale empirical measurements and learning.

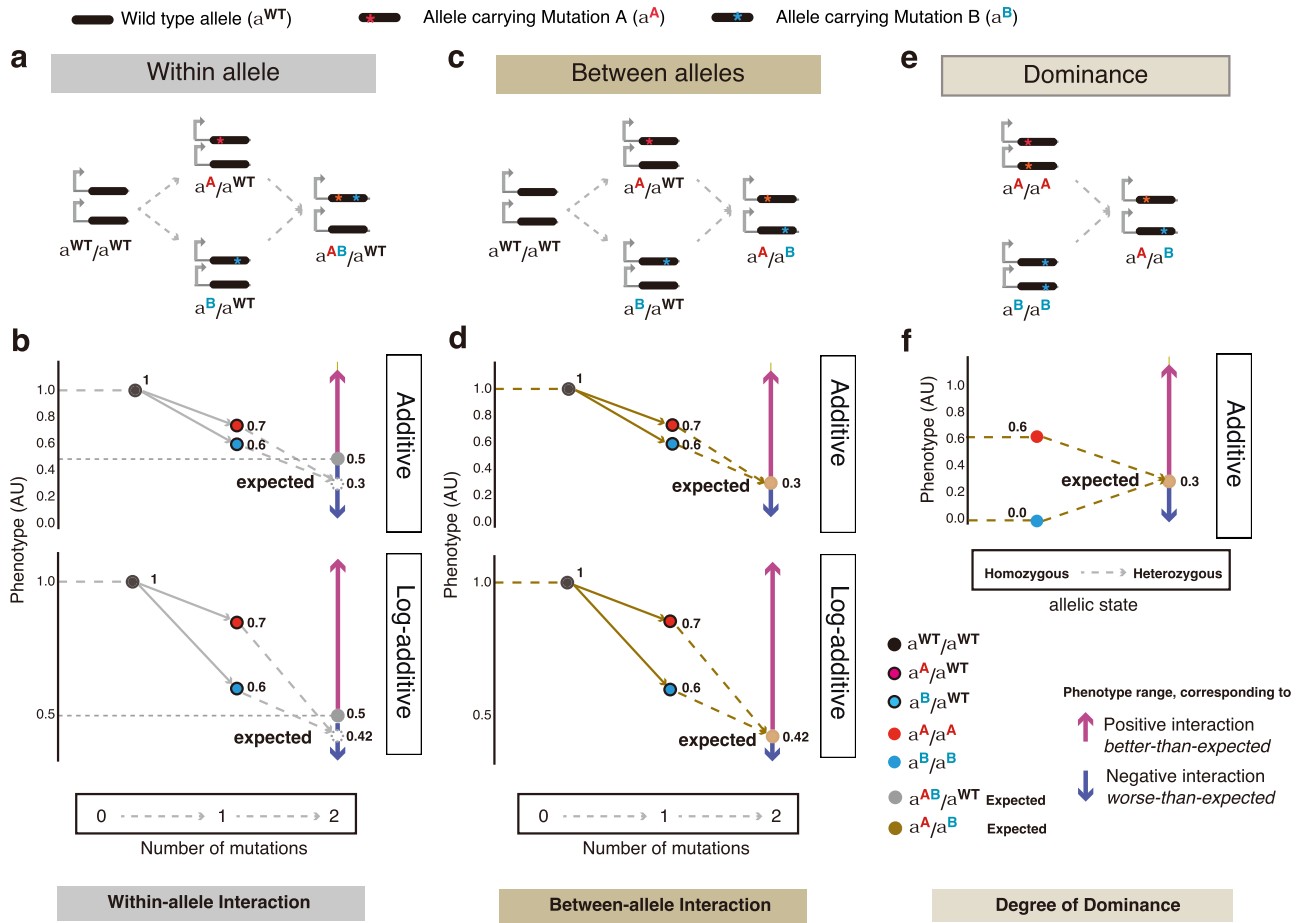

**Fig. 1 | Quantifying within-allele, between-allele genetic interactions and dominance. a–d** Two single mutations each on a chromosome of the same gene can combine within the same allele (**a**), quantified as within-allele interactions (intra-molecular epistasis) (**b**) or combine between two different alleles of the same gene (**c**), quantified as between-allele interactions (**d**), based on the additive or log-additive expectations (**b, d**). For within-allele mutation combinations, the lower bound of the expected phenotype is set to 0.5 AU (arbitrary unit), with the dotted grey empty circle indicating the expected phenotype below 0.5 AU and the solid grey circle indicating the new expected phenotype set to 0.5 AU. **e, f** How two homozygous mutations combine (**e**) is quantified as dominance based on the additive expectation (**f**).

## Results

### Within- and between-allele genetic interactions in simple biophysical systems

In a diploid system, each gene is present in two copies – in humans, the maternal and paternal alleles, respectively. When a gene carries two different mutations (mutant *A* and mutant *B* thereafter *A* and *B*), the two variants can, therefore, either both be present in the same allele ($\alpha^{AB}/\alpha^{WT}$) (Fig. 1a) or each variant can be in a different allele as a heteroallelic combination or compound heterozygote ($\alpha^{A}/\alpha^{B}$) (Fig. 1b). In a haploid organism, a similar comparison can be made for gene duplicates – two variants can be combined in the same duplicate (paralog) or each can be in a different paralog.

Quantifying how variants interact when they are combined requires the specification of a null model for independent effects. The interactions between variants within the same allele of a gene are typically referred to as epistasis or genetic interactions with a log-additive (or sometimes additive) null model most often used as the expected outcome (Eqs. 1–4, Fig. 1b)[1,4]. In contrast, interactions between variants in different alleles of the same gene are typically quantified as dominance, which uses an additive expected outcome (Eq. 5, Fig. 1e, f and Supplementary Fig. 1)[6,9]. Therefore, when comparing how mutations interact within and between alleles of a gene, throughout this manuscript we quantify interactions using both log-additive and additive null models (Eqs. 7, 8) (Fig. 1b, d).

Expected phenotypes ($W_{exp}$):

$$\log\left(W_{exp\_log}\right) = \text{Max}\begin{pmatrix} \log(C), \\ \log\left(W_{\alpha^{A}/\alpha^{WT}}\right) + \log\left(W_{\alpha^{B}/\alpha^{WT}}\right) - \log\left(W_{\alpha^{WT}/\alpha^{WT}}\right) \end{pmatrix}$$

(7)

$$W_{exp\_add} = \text{Max}\left(C, \left(W_{\alpha^{A}/\alpha^{WT}} + W_{\alpha^{B}/\alpha^{WT}} - W_{\alpha^{WT}/\alpha^{WT}}\right)\right)$$

(8)

$$C = \begin{cases} 0.5 \leftarrow \alpha^{AB}/\alpha^{WT} \\ 0 \leftarrow \alpha^{A}/\alpha^{B} \end{cases}$$

The wild-type phenotype $W_{\alpha^{WT}/\alpha^{WT}}$ is set to 1 and complete-loss of function as 0. For within-allele combinations of mutations ($\alpha^{AB}/\alpha^{WT}$), we set the lower bound of a phenotype to be half of the wild-type phenotype ($C = 0.5$) as the second allele of the gene is always functional (Fig. 1b) (Eqs. 7, 8), which is not a general definition but a reasonable treatment for our biophysical model.

The interaction between each pair of mutations was then quantified as the difference between the observed double mutant phenotype and that expected from an additive or log-additive model (Fig. 1b, d). Thus, a negative interaction score indicates a worse-than-expected phenotype when two mutations combine and vice versa for a positive interaction score.

For comparison, we also quantify the degree of dominance (thereafter, dominance; Eq. 6) by comparing the compound heterozygous phenotypes with the average of homozygotes' phenotypes (Eq. 5).

## Protein folding generates intra-molecular epistasis but not dominance

We first considered the simplest biophysical system where a phenotype (or fitness) is linearly related to the concentration of a protein that folds cooperatively and exists in two states: unfolded and folded (Model 1) (Fig. 2a, b, n). Such two-state cooperative folding is observed for many small proteins[30]. The probability of the protein being folded (or folded fraction, F) depends on the Gibbs free energy difference between the two states (the folding energy), $\Delta G_{\text{Folding}}$ (Fig. 2a, c) and can be calculated using the Boltzmann distribution (Eq. 9) where $R$ is the gas constant and $T$ is the absolute temperature.

$$F = \frac{e^{-\Delta G_{\text{Folding}}/RT}}{1 + e^{-\Delta G_{\text{Folding}}/RT}} \qquad (9)$$

Mutations affect the folding energy relative to the wild type, and changes in free energy are ($\Delta\Delta G_{\text{Folding}}$) assumed to be additive when combining mutations in the same molecule. We first considered a moderately stable protein ($\Delta G_{\text{Folding}} = -2$ kcal per mol) and mutations with a range of effect sizes ($\Delta\Delta G_{\text{Folding}} = -2$ to 13 kcal per mol) and combined pairs of mutations either within the same allele or between the two alleles (for details, see Methods).

We plotted the phenotypes of double mutants against the change in free energy ($\Delta\Delta G_{\text{Folding}}$) of the constituent single mutants (Fig. 2c–e, h, i) and also against the phenotypes of the single mutants (Fig. 2f, g, j, k). Comparing the iso-phenotype contour lines (phenotype isochores) when combining mutations within allele (Fig. 2d) to the expectations when assuming additivity (Fig. 2e upper panel) or log-additivity (Fig. 2e lower panel) illustrates how protein folding alone generates epistasis[1,13,15,16]: the sigmoidal relationship between the fraction of a protein that is folded and free energy (Fig. 2c) means that two destabilizing mutations often have an outcome that is more detrimental than both the additive and log-additive expectations, and combining a stabilizing with a destabilizing mutation often results in a better than expected outcome (negative and positive interactions respectively shown in Fig. 2e, g, Supplementary Fig. 2b).

In contrast, when two mutations affecting folding are combined in different alleles, the phenotypic outcome is additive (Fig. 2h–k, m), consistent with the most widely used null model for dominance (Fig. 2n–q). In comparison, assuming log-additivity overestimates the phenotype when combining two destabilizing mutations, resulting in negative interactions (lower panels of Fig. 2i, k). Considering more stable and less stable wild-type proteins does not change these conclusions (Supplementary Fig. 2a, c–f).

To evaluate these results experimentally, we expressed two copies of the N-terminal domain of the bacteriophage lambda repressor CI fused to GFP (Supplementary Fig. 3a, b Supplementary Fig. 9). This protein domain does not dimerize and forms a stable monomer following the two-state folding model[31]. We selected eight individual mutants of CI from our earlier study[21] (see the Methods section for more details) and quantified the total protein fluorescence when the mutations are combined in two different copies or in the same copy of the gene (Supplementary Fig. 3c, d). In agreement with the expectation from our simulations, mutations typically combine additively between alleles but have lower than expected concentrations when combined within the same allele (Supplementary Fig. 3c, d).

In summary, protein folding alone is not expected to generate between-allele interactions or dominance but it does generate within-allele interactions (epistasis): additivity is the correct null model for

dominance but neither additivity nor log-additivity is the correct expectation for epistasis.

## Ligand binding generates dominance

Many proteins bind ligands as part of their function, for example, small molecules, nucleic acids or other proteins. We examined how ligand-binding affects the expectation for how mutations interact within and between alleles. We first considered the case where a phenotype is linearly determined by the concentration of protein-ligand complex (Model 2) (Fig. 3a, b). In this system, mutations can alter the free energy of folding, as in Model 1 (Fig. 2c), or they can alter the binding energy (i.e. affect the binding affinity). In Figs. 3c and d, we compare the observed and expected double mutant phenotypes when combining mutations affecting either folding or binding (the expected values are the same for folding and binding mutations because the additive and log-additive expectations are calculated from the phenotypic values) In this model, the ligand concentration is the same as the protein concentration, the protein is moderately stable ($\Delta G_{\text{Folding}} = -2$ kcal per mol) and the binding affinity is moderate[32] (free energy of binding $\Delta G_{\text{Binding}} = -5$ kcal per mol, corresponding to a dissociation constant, $K_D = 291$ nM at 37 °C).

Similar to what is observed in the folding-only model (Model 1), when two destabilizing mutations are combined in the same protein, the decrease in the concentration of the protein bound to the ligand is often larger than the additive or log-additive expectation i.e., there is negative epistasis (blue shaded areas in Fig. 3c and Supplementary Fig. 4a, b).

However, unlike in Model 1, there are now non-additive interactions between mutations when they are combined in the two different alleles as heterozygotes, with two destabilizing mutations typically having an outcome that is better than the additive expectation (pink-shaded area indicating positive interaction scores in Fig. 3d, Fig. 3g and Supplementary Fig. 4c, d when [Ligand]/[Protein] = 1.0). Similarly, when two homozygous mutations combine (Fig. 3k), there is dominance (Fig. 3l–p). Weakly detrimental mutations combined with near-neutral mutations lead to worse-than-expected phenotypes (blue-shaded area in the top right of Fig. 3l) and better-than-expected phenotypes when combined with very detrimental mutations (pink-shaded area in the bottom left of Fig. 3l). The weakly detrimental mutations are partially dominant over mutations with near-neutral or very detrimental mutations. In short, mutations that do not show any dominance in the protein folding-only model now display dominance[22].

## Mutations interact with opposite signs within and between alleles

In the folding and binding model (Model 2), mutations interact to generate both within and between-allele interactions: the phenotype of double mutants often differs from the additive and log-additive expectations. However, comparing the curvature of the observed phenotypic isochores in Figs. 3c and d, the interactions within and between alleles are qualitatively different: two detrimental variants typically have an outcome that is worse than additive when they both occur in the same allele (Fig. 3c, e) but better than additive when they occur in different alleles (Fig. 3d, g). Moreover, although the signs of interaction are different, the interaction scores when the same mutations are combined within versus between alleles are positively correlated (Fig. 3i, j), meaning that mutation pairs with strong between-allele interactions show weak within-allele interactions and vice versa.

## Folding and binding mutants differ in their epistasis but have the same dominance

When mutations are combined in the same allele in Model 2, the outcome differs depending on whether the mutations affect folding or binding. That is, combining single mutants with the same

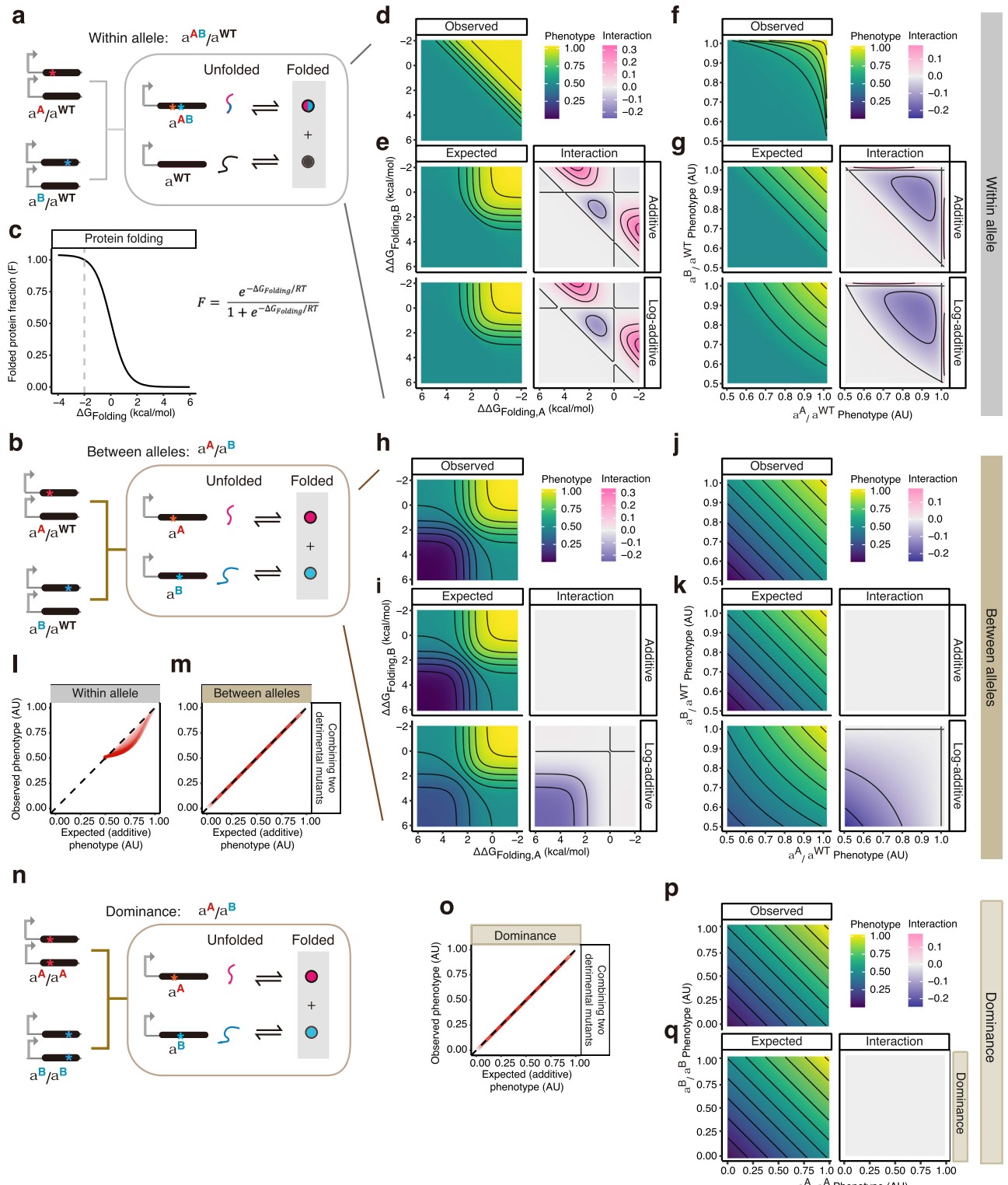

**Fig. 2 | Protein folding generates intra-molecular epistasis but not dominance.**
**a**, **b** Two-state protein folding system (Model 1) with two mutations within- (**a**) or between alleles (**b**). Phenotypes are determined by the folded protein concentration marked with grey-shaded boxes. **c** The relationship between the free energy changes of protein folding and folded protein fraction of homozygotes. The grey dashed line marks the wild-type protein free energy of folding. **d–k** Heatmaps show how two mutations combine within- (**d–g**) or between alleles (**h–k**) when they are ordered by free energy changes (**d**, **e**, **h**, **i**) or phenotypes (**f**, **g**, **j**, **k**). Black lines indicate phenotypic iso-chores. **l**, **m** Relationships between the observed and

expected phenotypes with additive expectation when combining two detrimental mutants within- (**l**) or between alleles (**m**). The darker the colour, the higher the density of the simulated data points. **n** Compound heterozygotes derived from two homozygous mutations within a two-state protein folding system (Model 1).
**o** Relationships between the observed and expected phenotypes with additive expectation when combining two detrimental homozygous mutants.
**p**, **q** Heatmaps show how two homozygous mutations combine when they are ordered by phenotypes either calculated (**p**) or expected based on the phenotype additivity (**q**). Black lines indicate phenotypic iso-chores.

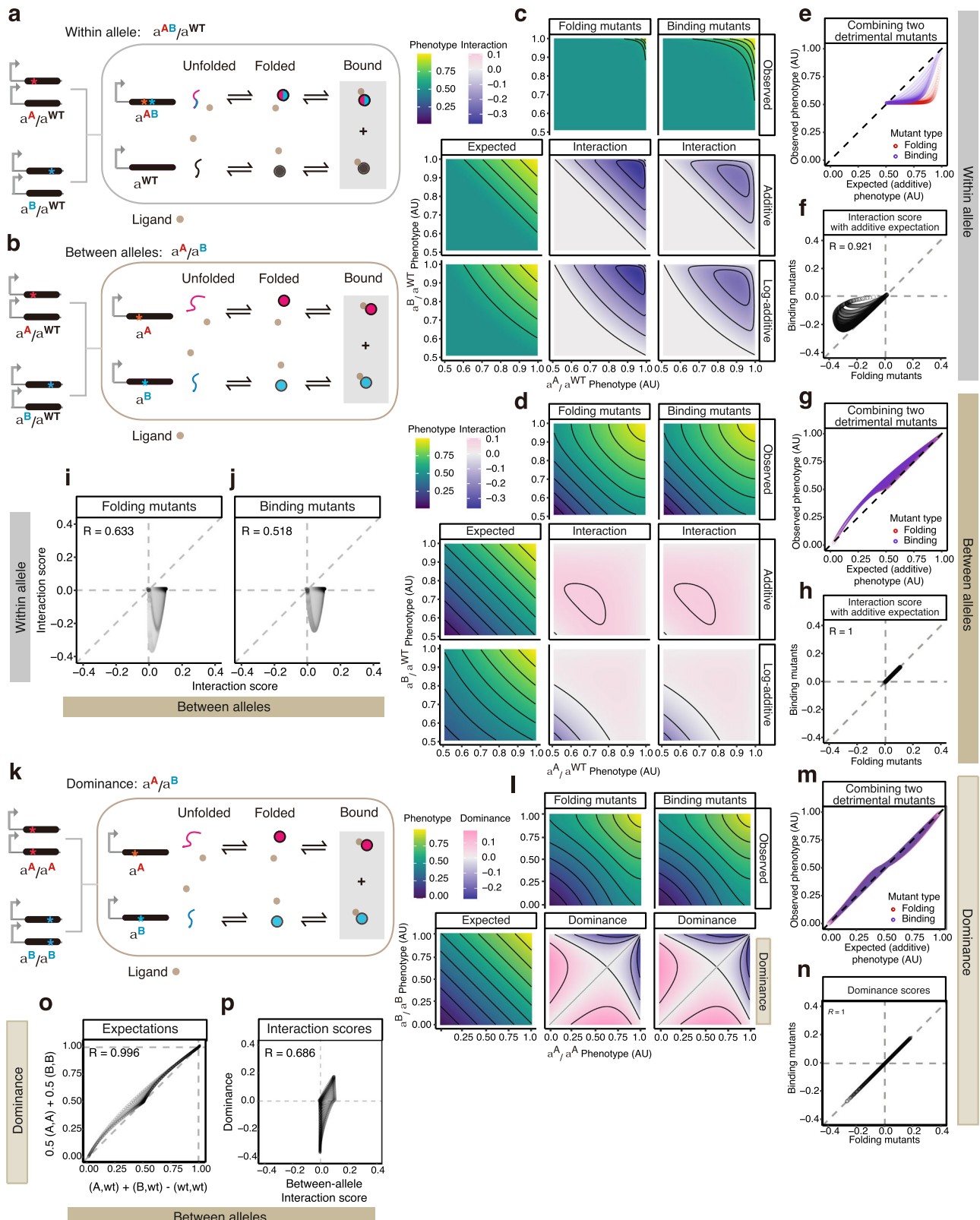

phenotypic value but different underlying causal biophysical mechanisms results in a different outcome in a double mutant (compare "Folding mutants" and "Binding mutants" columns in Fig. 3c). As shown in Fig. 3e, folding mutant phenotypes deviate from the expected additive phenotypes more than binding mutant phenotypes, resulting in stronger negative interaction scores (Fig. 3f). This is because the non-linear relationships between the

phenotype and the free energies of folding or binding are different (Supplementary Fig. 4a, b)[18,27,29].

In contrast, when two mutations in different alleles are combined, the phenotype of the double mutant does not depend on whether the mutations affect folding or binding (Fig. 3d, g, l, m), resulting in the same between-allele interactions (Fig. 3h) or dominance (Fig. 3n). Thus, in this simple folding and binding system, the magnitude of

**Fig. 3 | Ligand binding generates between-allele interactions and dominance.** **a**, **b** Three-state protein system with unfolded, folded, and ligand-bound states (Model 2), with two mutations of the same gene within- (**a**) or between alleles (**b**). Phenotypes are determined by the ligand-bound protein concentration marked with grey-shaded boxes. **c**, **d** Heatmaps showing how two mutations both affect the same biophysical parameters combine: protein-folding (the second column) or ligand-binding (the third column) when they are ordered by the phenotype. **e**, **g** Relationships between the observed and expected phenotypes with additive expectation when combining two detrimental mutants within (**e**) or between alleles (**g**). **f**, **h**–**j** Comparisons of interaction scores between different types of double mutant combinations: protein-folding vs. ligand-binding mutants within- (**f**) or between-alleles (**h**), between- vs. within-allele interactions of the protein-folding (**i**) or ligand-binding (**j**) mutants. The darker the colour, the higher the density of the simulated data points at the given between- vs. within-allele interactions. **k** Compound heterozygotes derived from two homozygous mutations within a three-state protein system (Model 2). **l** Heatmaps showing how two mutations both affect the same biophysical parameters combine: protein-folding (the second column) or ligand-binding (the third column) when they are ordered by the homozygous phenotype. **m** Relationships between the observed and expected phenotypes when combining two detrimental homozygous mutants. **n** Comparison of dominance between different types of double mutant combinations: protein-folding vs. ligand-binding mutants. **o**, **p** Comparisons of expected phenotypes (**o**) and dominance vs. between-allele interaction scores (**p**) for the same compound heterozygote mutants. The darker the colour, the higher the density of the simulated data points.

epistasis but not dominance depends on the biophysical effect of a mutation – whether a mutation affects protein stability or the binding affinity.

## Changes in ligand concentration alter both epistasis and dominance

In the version of Model 2 presented in Fig. 3, the ligand concentration is the same as the total protein concentration. However, as illustrated in Fig. 4, altering the ligand concentration both quantitatively and qualitatively changes epistasis, between-allele interactions, and dominance.

The interactions between mutations within the same allele change as the ligand concentration is altered (Fig. 4a) with the appearance of positive epistasis at low ligand concentrations (seen as the appearance of a magenta-filled area indicating positively interacting pairs in Fig. 4d, e). For example, when the ligand-protein ratio is 0.8, 80% of the protein is bound when both alleles are wild-type but 50% of the protein is bound when one allele is null. The lowest possible phenotypic value is thus 62.5% of the wild-type phenotype (0.5/0.8) not 50% of the wild-type as the additive and log-additive models assume (the left panel of Fig. 4d, e). To sum up, variations in the ligand concentration alter epistasis quantitatively when two mildly detrimental mutations combine (Fig. 4d, e, as the blue-shaded areas indicate) and qualitatively when two very detrimental mutations combine (Fig. 4d, e), seen as the change from positive epistasis to no epistasis with increasing ligand concentration indicated by the relative position of the expected versus observed phenotypes.

At high ligand concentrations (e.g. [Ligand]: [Protein] = 10 in Fig. 4b, c), there is no between-allele interaction or dominance, with mutations combining additively. When the ligand concentration is reduced (e.g. [Ligand]: [Protein] = 2 in Fig. 4b), the phenotype of a double mutant (compound heterozygote) is better than the additive expectation for between-allele interactions. There is also dominance, both positive and negative, depending upon the variant effect sizes (Fig. 4c, g). At [Ligand]: [Protein] = 1, these tendencies further increase (Fig. 4b, f). However, as the ligand concentration is further decreased, the phenotypes of double mutants become worse than the additive expectation for between-allele interactions ([Ligand]: [Protein] = 0.8 in Fig. 4b, f) and there is a positive dominance score ([Ligand]: [Protein] = 0.8 in Fig. 4c, g).

Thus, changing the concentration of a ligand both quantitatively and qualitatively alters the interactions between alleles, and can result in the more detrimental variant switching from dominant to recessive (Supplementary Fig. 5).

Why does ligand-binding cause between-allele interactions and dominance and why do the interactions switch as the ligand concentration changes? When the concentration of the ligand is in excess, changes in protein concentration result in a proportional change in ligand binding and the two alleles effectively behave as independent thermodynamic systems (Supplementary Fig. 6 right columns). However, when the ligand concentration is reduced, the two alleles now compete for binding to the ligand and so they can no longer be considered as independent systems. For example, when a mutation A destabilizes allele 1, less of allele 1 binds to the ligand but, because ligand binding is competitive, more of the ligand now binds to allele 2, resulting in a smaller-than-additive reduction in the total protein-ligand complex (see the magenta lines indicating additive expectation below the upper boundary of the darkest filled areas for the observed phenotypes in the middle columns Supplementary Fig. 6a, b). However, as the ligand concentration is further reduced, the system enters a regime where the relationship between the fraction bound and the total protein concentration is no longer linear as the protein is in excess and all of the ligand is bound (Supplementary Fig. 6). Moderately reduced stability or affinity now has no effect on the concentration of the protein-ligand complex such that only larger changes in energy alter the concentration of the bound complex (the left columns of Supplementary Fig. 6). As a result, many detrimental mutations combine to have a greater reduction in the protein-ligand complex than additive effects (see the magenta line above the upper boundary of the darkest filled area in the left column of Supplementary Fig. 6a, b).

## Nonlinear dose-response curves differentially transform within- and between-allele genetic interactions

So far, we have considered situations where a phenotype is linearly dependent on the concentration of a folded protein or a protein-ligand complex. However, in reality, phenotypes often depend non-linearly on the concentrations of macromolecules[33]. We, therefore, used three representative concentration-phenotype relationships – concave, convex and sigmoidal linking functions – to explore how non-linear relationships alter the interactions within and between alleles (Fig. 5a).

When applied to the mutations affecting ligand-binding energy (Model 2), all three functions alter within- and between-allele genetic interactions (Fig. 5b–d) and dominance (Fig. 5f, h). Moreover, the interactions not only change quantitatively but also in some cases qualitatively, switching from positive to negative (Fig. 5d, h). The effects of the non-linear concentration fitness functions can also differ for both interactions and dominance. For instance, a concave function shifts within-allele interactions and dominance to less negative or positive values, whereas between-allele interactions become negative (Fig. 5b, c, e–g). In contrast, a sigmoidal concentration-fitness function shifts both within- and between-allele interactions towards more negative values, with a stronger effect on the latter (Fig. 5d, e). Interestingly, a sigmoidal function switches dominance scores from negative to positive and vice versa, indicating mutations switch from recessive to dominant and vice versa (Fig. 5f, h). Moreover, between- and within-allele genetic interactions can become anti-correlated when nonlinear linking functions are applied (Fig. 5d, compared to Fig. 3i, j).

The effects of linking functions on the interactions between folding variants in Model 1 and Model 2 are consistent with those on binding mutants (Supplementary Fig. 7).

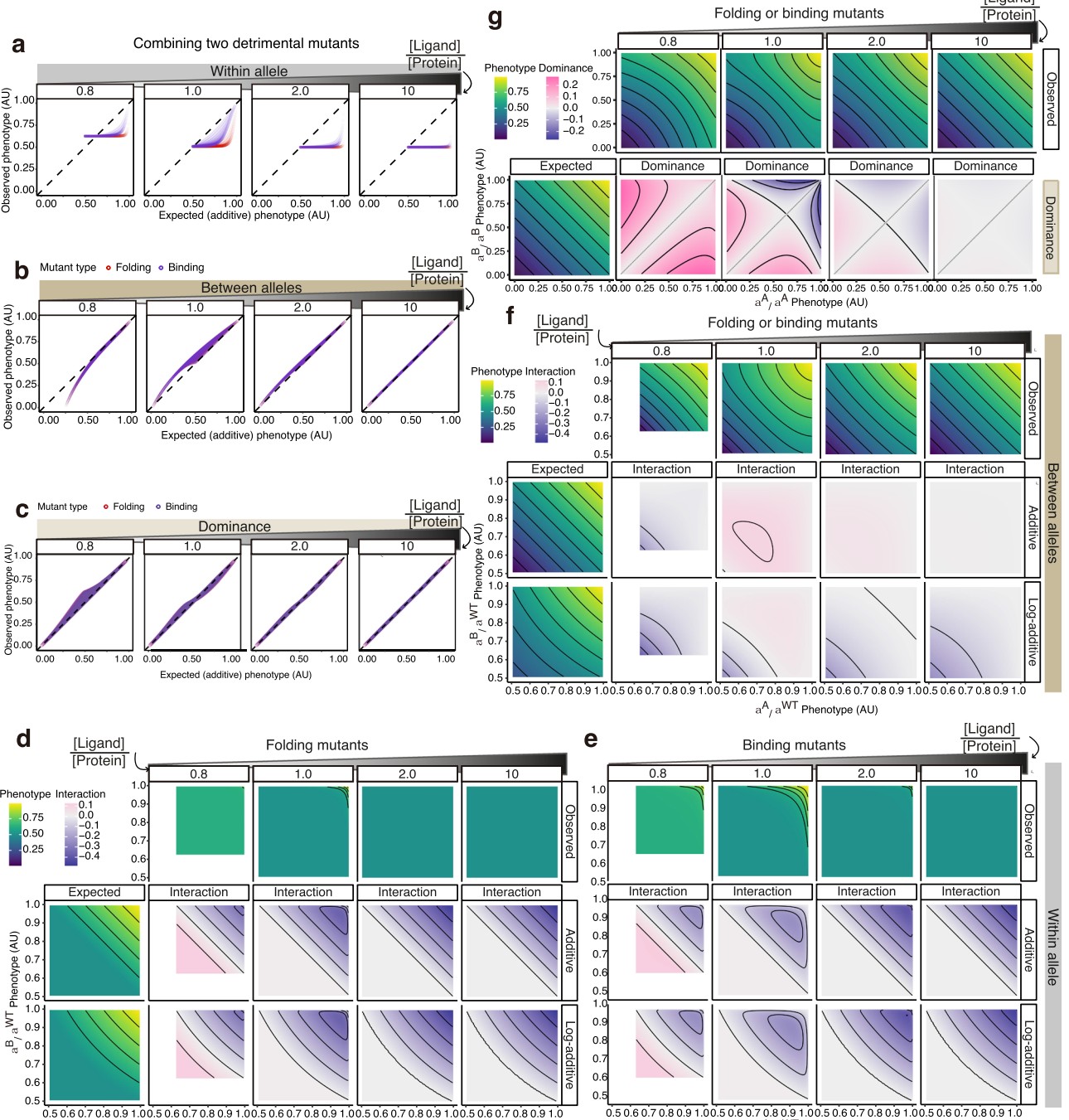

**Fig. 4 | Changes in ligand concentration switch between-allele interactions.** **a**, **b**, **c** The relationships between the observed and expected phenotypes with additive expectation when combining two detrimental mutants within- (**a**), between (**b**) heterozygous alleles or homozygous alleles (**c**) at different ligand-protein ratios. **d**–**g** Heatmaps show how two mutations combine within - (**d**, **e**) between heterozygous alleles (**f**), or homozygous alleles (**g**) when they both affect the same biophysical parameters: protein-folding (**d**, **f**, **g**) or ligand-binding (**e**, **f**, **g**). For both between-allele mutant combinations, folding or binding double mutants are shown together since they are not distinguishable (**f**, **g**). Black lines indicate phenotypic iso-chores.

To examine how dependent our conclusions are on the parameter values defining the non-linear curves, we repeated our analysis with five different curvatures (slopes) for each of the concave, convex and sigmoidal curves (Supplementary Fig. 8a). We observe stronger negative within-allele interactions with steeper convex curves and stronger negative between-allele interactions with steeper concave or sigmoidal curves (Supplementary Fig. 8b, d). Nevertheless, the direction of the changes is independent of the parameter values (Supplementary Fig. 8b–e).

Taken together, these results show that protein folding generates within-allele interactions (intra-molecular epistasis) and a single binding reaction is sufficient to generate between-allele interactions and dominance. However, the strength and sign of epistasis and dominance are highly dependent on the details of the system, including the relationship between the concentration of a molecule and the phenotype of interest. This makes them difficult to predict without a detailed mechanistic understanding of a system and knowledge of the relevant cellular parameters or large-scale empirical measurements of

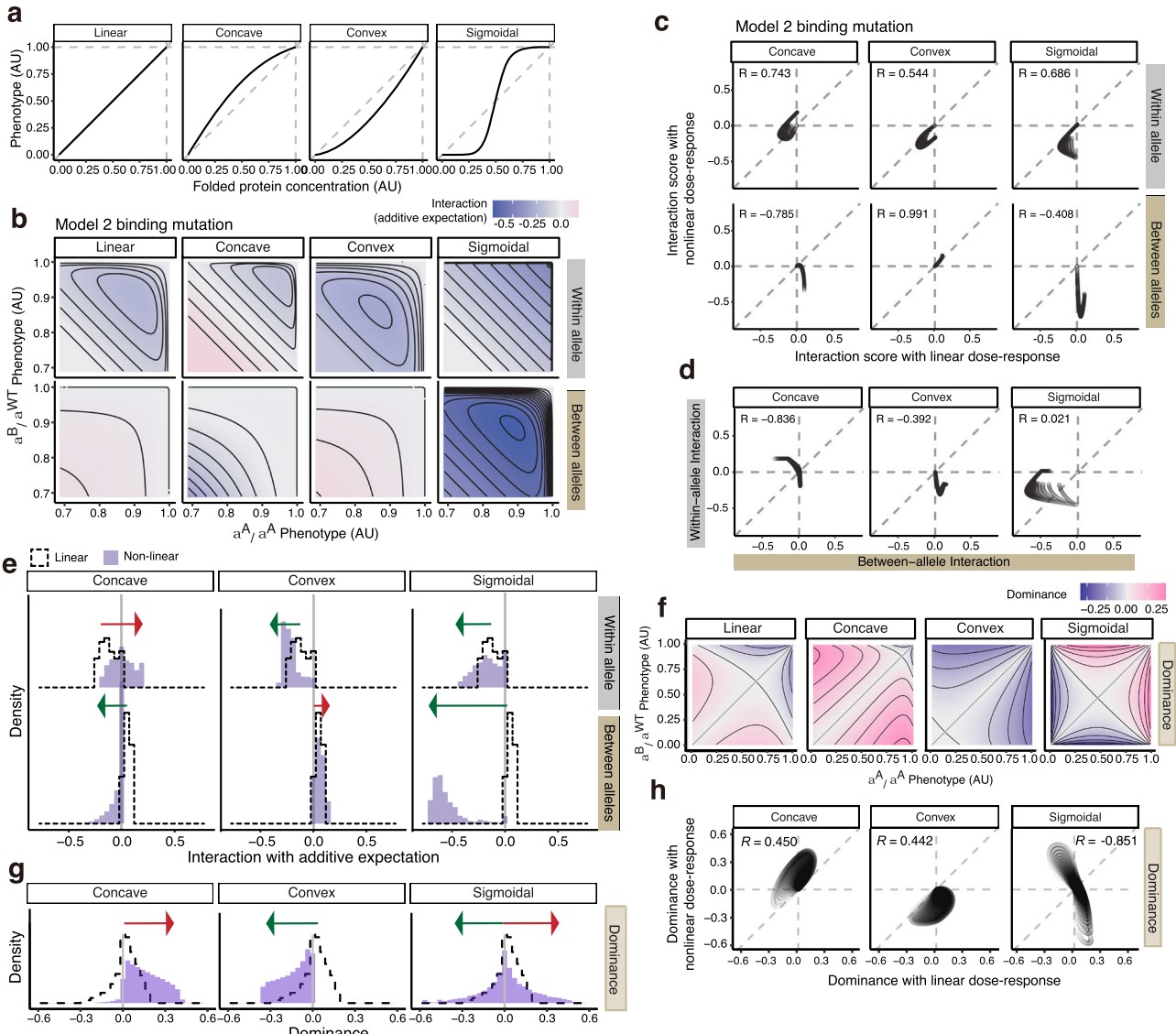

**Fig. 5 | Nonlinear concentration-phenotype functions differentially transform dominance and epistasis. a** Linear, concave, convex and sigmoidal linking functions are used to transform protein concentrations into phenotypes. **b** Interaction scores based on the additive expectation for double mutants within- or between-alleles with linear (Model 2), concave, convex, or sigmoidal protein concentration – phenotype relationships. **c** With vs. without nonlinear linking function comparisons of interaction scores based on the additive expectation. **d** Between- vs. within-allele double mutants' interaction scores based on the additive expectation, with nonlinear linking functions. **e** Distribution of interaction scores based on additive expectation before and after nonlinear linking functions. The green arrow indicates the distribution shifting towards negative values while the magenta arrow indicates the distribution shifting towards positive values; the arrowheads point at the range after applying the nonlinear linking functions to the phenotype. **f, g, h** Dominance scores before and after nonlinear linking functions (**f**), distribution (**g**), and with vs. without nonlinear linking function comparisons (**h**).

## Discussion

We have analyzed here how mutations interact within and between the two alleles of a gene in simple thermodynamic models of protein folding and binding in a one-gene diploid system. Our results show that protein folding on its own is expected to generate within-allele interactions (intra-molecular epistasis) but not between-allele interactions and dominance. However, the addition of a single binding reaction is sufficient to generate between-allele interactions and dominance.

How two mutations interact often differs in both magnitude and sign depending upon whether they combine within the same allele or in different alleles. Moreover, interactions depend qualitatively and quantitatively on both the biophysical effects of mutations and the context, with, for example, a change in the concentration of a ligand sufficient to switch a mutation from dominant to recessive in even the simple system of a protein that folds and binds a single ligand.

Taken together, our results illustrate that double mutant interactions are to be widely expected in even the simplest one-gene diploid biological systems, but also emphasize that they are system properties and therefore difficult to predict.

That dominance, between- and within-allele interactions are (1) expected in even the simplest systems, (2) context-dependent, and (3) difficult to predict – has important implications for human genetics, biotechnology, and evolution. As an example, our results suggest that it is not unreasonable to expect disease-causing alleles to switch from dominant to recessive depending upon the conditions, examined

mutational effects and interactions. Even the simple perturbation of altering the concentration of a binding partner can quantitatively alter epistasis and switch an allele from dominant to recessive.

phenotype or individual. Indeed, examples exist in human clinical genetics of variants reported to be both dominant and recessive, for example in Marfan syndrome[34].

Different and sometimes even opposite interaction patterns for the same pair of variants when they are combined on one allele rather than in two different alleles of a gene reinforces the importance of phasing variants into haplotypes in human genetics[35–37].

An important simplification of our approach is that it assumes that free energy changes are additive. Although this is likely to be true for the majority of mutation combinations[18,27] specific non-additive changes in free energy, for example between mutations in physically contacting residues[24,38], will generate additional epistasis and dominance.

In future work, it will be interesting to quantify and compare the expected within- and between-allele interactions in more complex equilibrium systems, as well as in dynamical models[8,39]. In addition, the effects of imbalanced expression of the different alleles could be combined with our model, as allele-specific expression has been widely observed in humans[35]. Finally, it will be important to experimentally evaluate dominance and epistasis in the same genes, for example, using deep mutational scanning approaches[1]. Using large-scale mutagenesis approaches it will also be possible to evaluate how frequently specific exceptions to the typical patterns of dominance and epistasis arise, what the most frequent causes of these exceptions are, and how they can be predicted.

An important general goal in human genetics, agricultural genetics, biotechnology and evolutionary biology including viral forecasting is to improve genetic prediction beyond the accuracy that can be achieved with additive models. Here we have shown that even in the very simplest biological systems, within, between-allele interactions and dominance are expected, context-dependent and challenging to predict. It is precisely the emergent, plastic and difficult-to-predict nature of these interactions that makes better-than-additive genetic prediction a formidable challenge.

How in practice will better-than-additive genetic prediction be achieved? One approach could be to build detailed mechanistic models of each system of interest and to parameterize these models using experimental measurements. However, given the complexity of most biological systems of interest, we suggest that a non-mechanistic approach may actually prove more successful. Indeed, we suspect that the combination of large-scale data collection and machine learning may prove to be the more efficient strategy for improving genetic prediction beyond what can be achieved with additive models and that this will accelerate progress in biotechnology, agriculture, viral forecasting, and human genetics.

## Methods
### Phenotypes
To study how mutations in a protein-coding gene combine in a diploid system, we first defined the phenotype ($W$) in the system as functional molecule concentration inside the cell relative to the wild-type diploid situation. These functional molecules can be folded proteins or a protein-ligand complex. In each model, the phenotype of a mutant is normalized to the homozygous wild-type (WT) phenotypes so that $W_{\alpha^{WT}/\alpha^{WT}} = 1$AU (arbitrary unit) and the complete loss of function is set to 0. With the simple assumption of additivity in the functional molecule concentrations, we imposed a lower bound of the expected simple heterozygote phenotype (i.e. any single mutants or any double mutants' expected phenotypes within the same allele) to 0.5 AU (half the wild-type phenotype) (Fig. 1b).

### Expected phenotypes, interaction scores and dominance scores
When there is only one mutation $A$, the homozygote is annotated as ($\alpha^A/\alpha^A$) and the heterozygote with the wild type is annotated as ($\alpha^A/$ $\alpha^{WT}$). The compound heterozygote is annotated as ($\alpha^A/\alpha^B$) while the heterozygous double mutant with both variants present in the same allele is designated ($\alpha^{AB}/\alpha^{WT}$). For a given pair of single mutant phenotypes, there are two expected double mutant phenotypes based on additive and log-additive expectations as shown in equations Eqs. (7, 8). For a given pair of homozygous mutant phenotypes, the expected double compound heterozygotes' phenotypes are given as the average of the parental phenotypes based on the widely used assumption for dominance calculation, as shown in equation Eq. (5). Interactions between mutations are quantified as differences between observed versus expected phenotypes as shown in Fig. 1.

### Model 1: protein folding
Phenotype is determined by the total concentration of folded protein. In this model, the protein of interest ($X$) expressed from each allele ($\alpha i$, $i \in \{1, 2\}$ – one maternal and the other paternal copy respectively, and the alleles are allowed to be the wild type) has two configuration states: unfolded ($X_{U,\alpha i}$) and folded ($X_{F,\alpha i}$). The free energy difference between folded and unfolded protein states is $\Delta G_{Folding,\alpha i}$ (kcal per mol). Mutations on each allele can affect folding energy ($\Delta G_{Folding}$), which is described as the sum of wild-type folding energy and the energy differences (mutations) $\Delta G_{Folding,wt} + \Delta\Delta G_{Folding,\alpha i}$. Equilibrium between the two states follows Eq. (10).

$$\frac{[X_{F,\alpha i}]}{[X_{U,\alpha i}]} = e^{-\left(\frac{\Delta G_{Folding,wt} + \Delta\Delta G_{Folding,\alpha i}}{RT}\right)} \tag{10}$$

In the above and following equations, $R$ is the gas constant ($R = 1.98 \times 10^{-3}$ kcal per mol), $T$ is the absolute temperature for 37 °C (310.15 Kelvin) and the wild-type $\Delta G_{Folding,wt}$ is set to −2 kcal per mol unless stated otherwise.

The total concentration of the protein ($X_T$) follows Eq. (11).

$$[X_T] = \sum_{i=1,2} \left([X_{U,\alpha i}] + [X_{F,\alpha i}]\right) \tag{11}$$

Expression levels from each allele were considered to be equal and therefore, $[X_{T,1}] = [X_{T,2}] = 0.5 [X_T]$ in all our models. Using Eqs. (10) and (11) with $[X_T]$ as a constant, we can calculate the functional molecule $[X_{F,\alpha i}]$ as a function of energy terms and total protein concentration in the following way:

$$[X_{F,\alpha i}] = \frac{0.5[X_T]}{1 + e^{\frac{\Delta G_{Folding,wt} + \Delta\Delta G_{Folding,\alpha i}}{RT}}} \tag{12}$$

The phenotype of a mutant $W_{mut}$ is the sum of the folded protein concentration normalized to that of the wild type, following Eq. (13):

$$W_{mut} = \frac{\sum_{i=1,2}[X_{F,\alpha i}]}{2[X_{F,wt}]} \tag{13}$$

To obtain the observed double mutant phenotypes between alleles, we could combine Eqs. (10–13) and simply replace $\Delta\Delta G_{Folding,\alpha 1}$ with $\Delta\Delta G_{Folding,A}$ and $\Delta\Delta G_{Folding,\alpha 2}$ with $\Delta\Delta G_{Folding,B}$ respectively, as shown below.

$$W_{\alpha^A/\alpha^B} = \frac{\left(1 + e^{\frac{\Delta G_{Folding,wt}}{RT}}\right)\left(2 + e^{\frac{\Delta G_{Folding,wt} + \Delta\Delta G_{Folding,A}}{RT}} + e^{\frac{\Delta G_{Folding,wt} + \Delta\Delta G_{Folding,B}}{RT}}\right)}{2\left(1 + e^{\frac{\Delta G_{Folding,wt} + \Delta\Delta G_{Folding,A}}{RT}}\right)\left(1 + e^{\frac{\Delta G_{Folding,wt} + \Delta\Delta G_{Folding,B}}{RT}}\right)} \tag{14}$$

For a single mutation $A$, we simply set $\Delta\Delta G_{Folding,\alpha 2} = 0$ and replace $\Delta\Delta G_{Folding,\alpha 1}$ with $\Delta\Delta G_{Folding,A}$ in Eq. (14). After simplification,

the equation for a single mutant phenotype can be written as follows:

$$W_{\alpha^A/\alpha^{WT}} = \frac{2 + e^{\frac{\Delta G_{\text{Folding,wt}}}{RT}} + e^{\frac{\Delta G_{\text{Folding,wt}} + \Delta\Delta G_{\text{Folding},A}}{RT}}}{2\left(1 + e^{\frac{\Delta G_{\text{Folding,wt}} + \Delta\Delta G_{\text{Folding},A}}{RT}}\right)} \tag{15}$$

For the double mutation (A and B) in the same molecule, $\Delta\Delta G_{\text{Folding},A}$ and $\Delta\Delta G_{\text{Folding},B}$ values are added to replace $\Delta\Delta G_{\text{Folding},A}$ in Eq. (15), resulting in Eq. (16).

$$W_{\alpha^{AB}/\alpha^{WT}} = \frac{2 + e^{\frac{\Delta G_{\text{Folding,wt}}}{RT}} + e^{\frac{\Delta G_{\text{Folding,wt}} + \Delta\Delta G_{\text{Folding},A} + \Delta\Delta G_{\text{Folding},B}}{RT}}}{2\left(1 + e^{\frac{\Delta G_{\text{Folding,wt}} + \Delta\Delta G_{\text{Folding},A} + \Delta\Delta G_{\text{Folding},B}}{RT}}\right)} \tag{16}$$

Additive or log-additive expectations of the double mutant phenotypes are calculated by combining Eq. (7) or Eq. (8) with Eq. (15) for both within-allele and between-allele combinations.

We note that the additive expectation for between-allele mutation combinations is mathematically the same as observed between-allele mutation combinations (Fig. 2i, k, upper right). Firstly, by combining Eq. (7) with Eq. (15) for each mutation, we obtain Eq. (17):

$$W_{\text{exp\_add}\_\alpha^A/\alpha^B} = \frac{2 + e^{\frac{\Delta G_{\text{Folding,wt}}}{RT}} + e^{\frac{\Delta G_{\text{Folding,wt}} + \Delta\Delta G_{\text{Folding},A}}{RT}}}{2\left(1 + e^{\frac{\Delta G_{\text{Folding,wt}} + \Delta\Delta G_{\text{Folding},A}}{RT}}\right)} + \frac{2 + e^{\frac{\Delta G_{\text{Folding,wt}}}{RT}} + e^{\frac{\Delta G_{\text{Folding,wt}} + \Delta\Delta G_{\text{Folding},B}}{RT}}}{2\left(1 + e^{\frac{\Delta G_{\text{Folding,wt}} + \Delta\Delta G_{\text{Folding},B}}{RT}}\right)} - 1 \tag{17}$$

Then, Eq. (17) can be rewritten into the same form as Eq. (14). In other words, mathematically $W_{\text{exp\_add}\_\alpha^A/\alpha^B} = W_{\alpha^A/\alpha^B}$ in this model. The log-additive expectation is, however, different.

By replacing mutation B in Eq. (14) with mutation A, we obtain the following formula, which is used to calculate the homozygous mutant's phenotype, as shown in (Eq.18) below.

$$W_{\alpha^A/\alpha^A} = \frac{1 + e^{\frac{\Delta G_{\text{Folding,wt}}}{RT}}}{1 + e^{\frac{\Delta G_{\text{Folding,wt}} + \Delta\Delta G_{\text{Folding},A}}{RT}}} \tag{18}$$

## Model 2: folding and ligand-binding

Phenotype is determined by the total concentration of protein-ligand complex in Model 2. In this model, one protein molecule binds to one ligand molecule and the total ligand concentration ($L_T$) is the sum of free ligand ($L$) and those bound to the protein (equal to the protein-ligand complex concentration, $\Sigma_{i=1,2}(X_{L,\alpha i})$), which follows Eq. (19).

$$[L_T] = [L] + \sum_{i=1,2}[X_{L,\alpha i}] \tag{19}$$

Compared to Model 1, there is an additional energy term – the free energy difference between the protein-ligand complex state and the unbound folded protein with free ligand state $\Delta G_{\text{Binding}}$ (kcal per mol) – which we define as binding energy. Mutations on each allele can now be described as those affecting protein folding energy ($\Delta\Delta G_{\text{Folding}}$) or binding energy ($\Delta\Delta G_{\text{Binding}}$).

There are three configuration states of the protein expressed from each allele: unfolded ($X_{U,\alpha i}$), folded ($X_{F,\alpha i}$) and protein-ligand complex ($X_{L,\alpha i}$). The equilibrium between $[X_{U,\alpha i}]$ and $[X_{F,\alpha i}]$ follows Eq. (10), and the equilibrium between $[X_{F,\alpha i}]$ and $[X_{L,\alpha i}]$ follows Eq. (20).

$$\frac{[X_{L,\alpha i}]}{[X_{F,\alpha i}] \cdot [L]} = e^{-\left(\frac{\Delta G_{\text{Binding,wt}} + \Delta\Delta G_{\text{Binding},\alpha i}}{RT}\right)} \tag{20}$$

The total concentration of the protein ($X_T$) follows Eq. (21), as shown below.

$$[X_T] = \sum_{i=1,2}\left([X_{U,\alpha i}] + [X_{F,\alpha i}] + [X_{L,\alpha i}]\right) \tag{21}$$

As stated earlier, $[X_{T,\alpha 1}] = [X_{T,\alpha 2}] = 0.5\,[X_T]$. By combining this information with Eq. (14), Eq. (19–21), we express the protein-ligand complex concentration from each allele as a function of energy terms, total protein concentration and the free ligand concentration as shown in Eq. (22):

$$[X_{L,\alpha i}] = \frac{0.5[X_T][L]}{[L] + e^{\frac{\Delta G_{\text{Binding,wt}} + \Delta\Delta G_{\text{Binding},\alpha i}}{RT}}\left(1 + e^{\frac{\Delta G_{\text{Folding,wt}} + \Delta\Delta G_{\text{Folding},\alpha i}}{RT}}\right)} \tag{22}$$

The phenotype of a mutant $W_{\text{mut}}$ is the sum of the protein-ligand complex concentration normalized to that of the wild-type as shown below in Eq. (23).

$$W_{\text{mut}} = \frac{\sum_{i=1,2}[X_{L,\alpha i}]}{2[X_{L,\text{wt}}]} \tag{23}$$

For a double mutant (A and B mutations respectively) between alleles, we combine Eqs. (10, 19–23) and simply replace $\alpha 1$ with A and $\alpha 2$ with B for corresponding mutation types, as shown in Eq. (24).

$$W_{\alpha^A/\alpha^B} = \frac{[L]\left([L_{\text{wt}}] + e^{\frac{\Delta G_{\text{Binding,wt}}}{RT}}\left(1 + e^{\frac{\Delta G_{\text{Folding,wt}}}{RT}}\right)\right)}{2[L_{\text{wt}}]\left([L] + e^{\frac{\Delta G_{\text{Binding,wt}} + \Delta\Delta G_{\text{Binding},A}}{RT}}\left(1 + e^{\frac{\Delta G_{\text{Folding,wt}} + \Delta\Delta G_{\text{Folding},A}}{RT}}\right)\right)}$$
$$\bullet \frac{\left(2[L] + e^{\frac{\Delta G_{\text{Binding,wt}} + \Delta\Delta G_{\text{Binding},A}}{RT}}\left(1 + e^{\frac{\Delta G_{\text{Folding,wt}} + \Delta\Delta G_{\text{Folding},A}}{RT}}\right) + e^{\frac{\Delta G_{\text{Binding,wt}} + \Delta\Delta G_{\text{Binding},B}}{RT}}\left(1 + e^{\frac{\Delta G_{\text{Folding,wt}} + \Delta\Delta G_{\text{Folding},B}}{RT}}\right)\right)}{\left([L] + e^{\frac{\Delta G_{\text{Binding,wt}} + \Delta\Delta G_{\text{Binding},B}}{RT}}\left(1 + e^{\frac{\Delta G_{\text{Folding,wt}} + \Delta\Delta G_{\text{Folding},B}}{RT}}\right)\right)} \tag{24}$$

If there is only one mutation A on allele 1, then $\Delta\Delta G_{\text{Folding},\alpha 2} = \Delta\Delta G_{\text{Binding},\alpha 2} = 0$ and we replace $\Delta\Delta G_{\text{Folding},\alpha 1}$ with $\Delta\Delta G_{\text{Folding},A}$ and $\Delta\Delta G_{\text{Binding},\alpha 1}$ with $\Delta\Delta G_{\text{Binding},A}$. Eq. (24) will be simplified to a single mutant phenotype as follows:

$$W_{\alpha^A/\alpha^{WT}} = \frac{[L]\left([L_{\text{wt}}] + e^{\frac{\Delta G_{\text{Binding,wt}}}{RT}}\left(1 + e^{\frac{\Delta G_{\text{Folding,wt}}}{RT}}\right)\right)}{2[L_{\text{wt}}]\left([L] + e^{\frac{\Delta G_{\text{Binding,wt}} + \Delta\Delta G_{\text{Binding},A}}{RT}}\left(1 + e^{\frac{\Delta G_{\text{Folding,wt}} + \Delta\Delta G_{\text{Folding},A}}{RT}}\right)\right)}$$
$$\bullet \frac{\left(2[L] + e^{\frac{\Delta G_{\text{Binding,wt}} + \Delta\Delta G_{\text{Binding},A}}{RT}}\left(1 + e^{\frac{\Delta G_{\text{Folding,wt}} + \Delta\Delta G_{\text{Folding},A}}{RT}}\right) + e^{\frac{\Delta G_{\text{Binding,wt}}}{RT}}\left(1 + e^{\frac{\Delta G_{\text{Folding,wt}}}{RT}}\right)\right)}{\left([L] + e^{\frac{\Delta G_{\text{Binding,wt}}}{RT}}\left(1 + e^{\frac{\Delta G_{\text{Folding,wt}}}{RT}}\right)\right)} \tag{25}$$

Above, $[L_{\text{wt}}]$ is the free ligand concentration inside the wild type while $[L]$ is the free ligand concentration inside the mutant cells. For a double mutant on the same allele, $\Delta\Delta G_{\text{Folding},A}$ and $\Delta\Delta G_{\text{Folding},B}$ are summed in place of $\Delta\Delta G_{\text{Folding},A}$, and $\Delta\Delta G_{\text{Binding},A}$ and $\Delta\Delta G_{\text{Binding},B}$ are summed in place of $\Delta\Delta G_{\text{Binding},A}$ while keeping allele 2 as wild-type ($\Delta\Delta G_{\text{Folding},\alpha 2} = 0$ and $\Delta\Delta G_{\text{Binding},\alpha 2} = 0$), shown as in Eq. (26).

$$W_{\alpha^{AB}/\alpha^{WT}} = \frac{[L]\left([L_{\text{wt}}] + e^{\frac{\Delta G_{\text{Binding,wt}}}{RT}}\left(1 + e^{\frac{\Delta G_{\text{Folding,wt}}}{RT}}\right)\right)}{2[L_{\text{wt}}]\left([L] + e^{\frac{\Delta G_{\text{Binding,wt}}}{RT}}\left(1 + e^{\frac{\Delta G_{\text{Folding,wt}}}{RT}}\right)\right)}$$
$$\bullet \frac{\left(2[L] + e^{\frac{\Delta G_{\text{Binding,wt}} + \Delta\Delta G_{\text{Binding},A} + \Delta\Delta G_{\text{Binding},B}}{RT}}\left(1 + e^{\frac{\Delta G_{\text{Folding,wt}} + \Delta\Delta G_{\text{Folding},A} + \Delta\Delta G_{\text{Folding},B}}{RT}}\right) + e^{\frac{\Delta G_{\text{Binding,wt}}}{RT}}\left(1 + e^{\frac{\Delta G_{\text{Folding,wt}}}{RT}}\right)\right)}{\left([L] + e^{\frac{\Delta G_{\text{Binding,wt}} + \Delta\Delta G_{\text{Binding},A} + \Delta\Delta G_{\text{Binding},B}}{RT}}\left(1 + e^{\frac{\Delta G_{\text{Folding,wt}} + \Delta\Delta G_{\text{Folding},A} + \Delta\Delta G_{\text{Folding},B}}{RT}}\right)\right)} \tag{26}$$

## Table 1 | Parameter values

| | Mutation types | $\Delta G_{\text{Folding,wt}}$ (kcal per mol) | $\Delta G_{\text{Binding,wt}}$ (kcal per mol) | $L_T$ Ratio to $X_T$ |
|---|---|---|---|---|
| Model 1 | Folding | −2/−3.5/−0.5 | NA | NA |
| Model 2 | Folding, Binding | −2 | −5 | 1/0.8/2/10 |

*F* Folding. *L* Ligand-binding. "/" separates different parameter options that we examined in this study and NA indicates that the parameter is not relevant to the model.

Additive or log-additive expectation can be calculated by combining Eq. (7) or Eq. (8) with Eq. (25), with corresponding parameters changed in the equations.

By replacing mutation *B* in Eq. (24) with mutation *A*, we obtain the following formula to calculate a homozygous mutant's phenotype.

$$W_{\alpha^A/\alpha^A} = \frac{[L]\left([L_{\text{wt}}] + e^{\frac{\Delta G_{\text{Binding,wt}}}{RT}}\left(1 + e^{\frac{\Delta G_{\text{Folding,wt}}}{RT}}\right)\right)}{[L_{\text{wt}}]\left([L] + e^{\frac{\Delta G_{\text{Binding,wt}} + \Delta\Delta G_{\text{Binding,A}}}{RT}}\left(1 + e^{\frac{\Delta G_{\text{Folding,wt}} + \Delta\Delta G_{\text{Folding,A}}}{RT}}\right)\right)} \quad (27)$$

To be noted, the free ligand concentrations in the cell will change with the change of corresponding phenotypes. For example, the free ligand concentrations in single mutant *A* or *B* are $[L_A] = [L_T] - ([X_{L,A}] + [X_{L,\text{WT}}])$ or $[L_B] = [L_T] - ([X_{L,B}] + [X_{L,\text{WT}}])$. If $[L_T]$ is much bigger than $[X_T]$ and thus the free ligand concentration is not affected by how many molecules are bound to the protein, free ligand concentration can be considered a constant as $[L_T]$ (i.e. $[L_A] \approx [L_B] \approx [L_T]$). In this situation, the additive expectation for a between-allele mutant combination simplifies to the same form as Eq. (24), indicating that there will be no dominance when the total ligand concentration is much bigger than the total protein concentration.

We examined several cases where we altered $[L_T]$ so that the total ligand concentration is the same, more abundant (2X, 10X) or less abundant (0.8X) than the total protein concentration $[X_T]$. With mutational effects as changes in folding energy ($\Delta\Delta G_{\text{Folding}, \alpha i}$) or ligand-binding energy ($\Delta\Delta G_{\text{Binding}, \alpha i}$) as input, we calculated the phenotypes using the R package rootSolve.

### General non-linear functions

Three common types of non-linear functions (concave, convex, and sigmoidal) linking functional protein concentrations to the downstream phenotypes were used. All the curves go through points (0, 0) and (1, 1). To evaluate the parameter sensitivity of our conclusions, we generated another set of equations defining the curves with a tunable parameter *m*. In the equations below, *x* and *y* define phenotypes before and after applying each of the non-linear transformations.

Concave:

$$y = \frac{e^{-m(x-1)} - e^m}{1 - e^m} \quad (28)$$

Convex:

$$y = \frac{e^{m(x-1)} - e^{-m}}{1 - e^{-m}} \quad (29)$$

Sigmoidal:

$$y = \frac{e^m + 1}{e^m - 1}\left(\frac{1}{1 + e^{-2m(x-0.5)}} - \frac{1}{1 + e^m}\right) \quad (30)$$

$m \geq 0$, and the bigger *m* is, the bigger the curvature is. When $m = 0$, it becomes a line. We examined five parameter values (0.6, 1.5, 2.5, 3.8, and 5.5 for concave and convex curves; 2.2, 3.5, 5, 7 and 10 for each sigmoidal curve) and their effects on the phenotypes and interactions

in our study. In Fig. 5, we selected $m = 1.5$ for concave and convex curves; $m = 5$ for the sigmoidal curve.

### Simulation of mutational effects

Wild-type biophysical parameters and relative ligand concentration to the total protein concentrations in each model are shown in Table 1.

Two mutation types – those affecting protein stability (folding mutations) and binding affinity to a ligand (binding mutations) were described as changes in Gibbs free energy ($\Delta\Delta G$) between folded-unfolded states ($\Delta\Delta G_{\text{Folding}}$) and bound-unbound states ($\Delta\Delta G_{\text{Binding}}$). To generate single mutations, $\Delta\Delta G_{\text{Folding}}$ and $\Delta\Delta G_{\text{Binding}}$ ranging from −2 to 13 kcal per mol with an interval of 0.125 kcal per mol were added to $\Delta G_{\text{Folding,wt}}$ and $\Delta G_{\text{Binding,wt}}$ respectively.

To examine how two mutations of given phenotypes combine, we generated single mutants with phenotypes ranging from 0.5 AU to 1.02 AU with an interval of 0.005 AU, and homozygous mutants with phenotypes ranging from 0 AU to 1.02 AU with an interval of 0.01 AU. Using the same sets of equations, and phenotypes as inputs, again using R and rootSolve package, we calculated the corresponding $\Delta\Delta G$ for each phenotype. Then, with the obtained $\Delta\Delta G$ values for each mutant as new inputs to the system, we calculated double mutants' phenotypes.

As stated earlier, when two mutations affecting the same biophysical parameters combined on the same allele, $\Delta\Delta G$ was added to the corresponding parameter as the new input to calculate double mutants' phenotypes. On the other hand, when mutations combined between alleles, $\Delta\Delta G$ was kept separate and inputted as two independent values on each allele for the phenotype calculation.

The processes of calculating phenotypes from mutants ($\Delta\Delta G$) and $\Delta\Delta G$ from phenotypes with the nonlinear linking functions are the same as those without linking curves, except that the phenotypes were transformed based on Eqs. (28–30) depending on the situation. The transformed single mutant phenotypes were used to calculate additive or log-additive expectations based on Eqs. (5, 7, 8), and interaction scores were calculated as shown in Fig. 1. All the phenotype and interaction patterns were visualized using the ggplot package in R.

### The 'diploid' plasmid constructs for Model 1

We generated plasmids carrying two copies of the N-terminal domain of the lambda repressor CI fused with GFP (Supplementary Fig. 3a, b). The 'diploid' plasmid carries two copies of CI N-terminal domain (1st −91st AA out of the 236AA) fused with GFP at the C-terminus via a flexible glycine linker (36 bp: ggatccgctggctccgctgctggttctggcgaattc) in the tail-to-tail direction separated by a bidirectional terminator present in the template plasmid. It has been shown in various studies that fluorescence intensity is linearly related to the fluorescent protein or the tagged protein[40–43], allowing us to examine the phenotype additive assumptions.

To generate the 'diploid' plasmid, we first constructed an intermediate 'haploid' plasmid carrying the fusion protein of CI N-terminal domain and GFP, generated from the template pCIPR-Low plasmid[21] which carries one copy of the full-length CI gene, operator DNA sequence and the downstream GFP ORF[21]. The 'haploid' plasmid shares the same backbone of the pCIPR-Low plasmid (PBADM11), but with the operator removed so that CI cannot dimerize or bind to any molecules.

The glycine linker was inserted via site-directed mutagenesis (SDM) using Vazyme Phanta Max Super-Fidelity DNA Polymerase

(primers are listed in Supplementary Table 3) followed by New England Biolabs (NEB) KLD enzyme treatment. We used the protocols of the NEB Q5SDM. Single mutations of CI are introduced to the 'haploid' plasmid via SDM and chemically transformed to the commercial *E.coli* DH5α competent cells (purchased from Genesand) following the manufacturer's instructions.

To generate the 'diploid' plasmid, the gene block region starting from the promoter to the stop codon of CI-GFP fusion protein was amplified via overlap extension PCR that introduces two restriction enzyme sites Spe1 and Sac1 (Supplementary Fig. 9). Then, the PCR amplicon is purified, and treated with restriction enzymes NEB Spe1-HF and Sac1-HF at 37 °C for 1 h. This fragment is ligated to the linearized WT 'haploid' plasmid sharing the same restriction site sticky ends via Vazyme T4 ligase treatment at room temperature for 1 h to form a diploid system (Supplementary Fig. 9a).

For the within-allele double mutants, two mutations instead of one are introduced to the 'haploid' plasmid via SDM as the method described above. The plasmid is linearized downstream to the terminator and ligated with the WT CI N terminal-GFP fusion protein ORF (Supplementary Fig. 9b). Combining two different gene copies each carrying one different mutation, we generated between-allele double mutations (Supplementary Fig. 9c).

With reference to all 'haploid' mutant phenotype data and structural characteristics of mutation sites[21], we selected 8 single mutations (Table 2, Supplementary Table 1). Mutational effects on the function of the full-length CI protein were highly bimodal with most mutations either neutral or completely detrimental. We reason that our model will unmask more mildly destabilizing mutations as there is no dimerization that is known to stabilize the protein. Therefore, the eight mutations were selected with the phenotype spanning from near neutral to mildly detrimental and with low standard error. We did not select any completely detrimental mutations as they are not informative when combined with other mutations. Mutational effects correlate well between our 'heterozygous diploid' protein-folding system and the earlier study, with this study more sensitive to mutational effects on protein stability (Supplementary Fig. 10). An exception is the mutant L33F (A99C) which is detrimental in the original data but neutral in this new model system, indicating that the mutation does not affect protein folding.

To generate mutation combinations, we excluded mutation pairs whose amino acid alpha-carbon atoms are within 12 Å distances in the protein 3D structure (PDB 1LMB) to exclude any potential position-specific interactions due to the contacts[44]. With several mutations failing during PCR or subsequent sequence-verification steps, we obtained in total 23 double mutations (10 within-allele combinations and 13 between-allele combinations) (Supplementary Table 2). Two of these double mutations are also observed in the earlier study based on the pCIPR-Low[21] (See the phenotype comparisons in Supplementary Figure 10, and Supplementary Table 4 for the full phenotype data).

### Phenotyping the mutational effects via Flow cytometry

The sequence-validated diploid plasmids were transformed into the homemade *E.coli* BW27783 competent cells for arabinose-induced fusion protein production. Three colonies were picked for each genotype as biological replicates and cultured in LB-ampicillin liquid media. 1.5 μl of the overnight LB bacteria culture was added to 150 μl M9 medium in 96-well plates and incubated for about 6 h till OD600 = 0.4–0.6, to induce the fusion protein expression. Each sample was prepared with 1:500 dilution into 2 ml PBS for flow cytometry. Samples were then analysed with BD FACSMelody with a 100-flow rate and 5000 cells are recorded based on SSC-A and FSC-A gate threshold in four batches (Supplementary Fig. 11). Cells carrying the empty plasmid backbone (PBADM11) were used as the GFP-negative control to count the autofluorescence. BD FACSChorus software version

**Table 2 | Single mutations were chosen from Li et al., 2019[21]**

| Substitution site (AA) | Substitution site (nt) | Mutation position at the structure |
|---|---|---|
| M23L | A67C | surface |
| L33F | A99C | DNA-binding |
| K53N | A159C | surface |
| S55R | C165G | core |
| V56I | G166A | core |
| E57K | G169A | surface |
| E58D | A174T | surface |
| F59L | T177A | core |

The position is indexed based on the Li et al. [21] study. The position in the full protein sequence is +17 aa.

1.1.20.2030 was used to analyze GFP signal of the cells during the data collection.

### Flow cytometry data analysis

We used R to subset cells with the threshold of FSC-A between 900 and 9800, SSC-A between 110 and lg(0.93)xFSC-A + 530. Bacterial cells were visibly in two different populations – with high or low SSC. To decrease the heterogeneity, we chose the bacterial cells with low SSC (low subcellular complexity) (Supplementary Fig. 11). The mean GFP signal of each mutant cell population was averaged across three biological replicates ($\mu\_GFP_{mut}$), removed the mean autofluorescence ($\mu\_GFP_{PBAD}$) and finally normalized to the wild type ($\mu\_GFP_{wt}$) within the same batch, following the formula shown in Eq. (31).

$$GFP_{obs} = \frac{\mu\_GFP_{mut} - \mu\_GFP_{PBAD}}{\mu\_GFP_{wt} - \mu\_GFP_{PBAD}} \tag{31}$$

The standard error of the mean (SE) of the observed normalized GFP signal follows Eq. (32), as shown below:

$$SE_{obs} = \sqrt{\left(\frac{\sqrt{SE_{mut}^2 + SE_{PBAD}^2}}{\mu\_GFP_{mut} - \mu\_GFP_{PBAD}}\right)^2 + \left(\frac{\sqrt{SE_{wt}^2 + SE_{PBAD}^2}}{\mu\_GFP_{wt} - \mu\_GFP_{PBAD}}\right)^2} \tag{32}$$

The expected phenotype of double mutants is calculated according to equations Eqs. (7, 8), with the error propagated using the formulas shown below:

Additive:

$$SE_{exp\_add\_AB} = \sqrt{SE_{obs\_A}^2 + SE_{obs\_B}^2 + SE_{obs\_wt}^2} \tag{33}$$

Log-additive:

$$SE_{exp\_log\_AB} = \sqrt{\left(\frac{SE_{obs\_A}}{GFP_{obs\_A}}\right)^2 + \left(\frac{SE_{obs\_B}}{GFP_{obs\_B}}\right)^2} \tag{34}$$

Interactions between or within alleles are calculated as shown in Fig. 1, and the error of the interaction scores is shown below.

$$SE_{interaction} = \sqrt{SE_{obs\_AB}^2 + SE_{exp\_AB}^2} \tag{35}$$

### Reporting summary

Further information on research design is available in the Nature Portfolio Reporting Summary linked to this article.

## Data availability

The experimental data generated to support the conclusions of this study are included in the article and the supplementary information. The datasets generated and analyzed in this study are also available in the GitHub repository: XL-Lab/P1_Dominance_vs_Epistasis[45].

## Code availability

Custom codes are available in the GitHub repository: https://github.com/XLi-Lab/P1_Dominance_vs_Epistasis [github.com][45].

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

## Acknowledgements

Work in the lab of X.L. was supported by the Young Scientists Fund of the National Natural Science Foundation of China (NSFC Grant No. 32100478 to X.L.). Work in the lab of B.L. was supported by the European Research Council (ERC, Advanced Grant 'Mutanomics' 883742 to B.L.), the la Caixa Research Foundation (LCF/PR/HR21/52410004 to B.L.), the Bettencourt Schueller Foundation, the AXA Research Foundation, the Spanish Ministry of Science, Innovation and Universities (PID2020-118723GB-I00 to B.L.), Agencia de Gestio d'Ajuts Universitaris i de Recerca (AGAUR, 2017 SGR 1322 to B.L.) and the CERCA Program/Generalitat de Catalunya. We also acknowledge the support of the Spanish Ministry of Science and Innovation to the EMBL partnership and the Centro de Excelencia Severo Ochoa.

## Author contributions

X.X. performed all analyses and made the figures with the assistance of X.L. X.X, X.S., and Y.W. performed experiments. X.L. and B.L. conceived the project, designed the analyses and wrote the manuscript with input from X.X.

## Competing interests

The authors declare no competing interests.
