## [Peer Review File · Nature Communications]

Dominance vs. epistasis: the biophysical origins and plasticity of genetic interactions within and between allelesREVIEWER COMMENTS

Reviewer #1 (Remarks to the Author):

Thank you for the opportunity to review “Dominance vs. epistasis: the biophysical origins and plasticity of genetic interactions within and between alleles” by Xie et al. This manuscript discusses how to predict phenotypes as a result of multiple mutations within and between alleles that lead to epistasis vs dominance. Authors use simulations of a simple biophysical system to examine mutations within and between alleles that affect folding or ligand binding. They show that the emergence of epistasis or dominance depends on the biophysical property of the mutation highlighting the difficulty in predicting phenotypes.

This study investigates how mutations combine to impact phenotypes and result in epistatic vs dominant effects. This addresses an important question in the field of genetic interactions and provides novel insight about how mutations that affect different biochemical properties combine to affect phenotypes. This is an interesting study although the manuscript is very difficult to read and does not provide a sufficient level of detail necessary to understand the results.

Specific comments:

Figure legends should explain what the different colours, outlines within the heatmaps and dashed lines mean for every panel. Currently, it is not clear what the figures show. For example, authors indicate that on lines 208-211 “Similar to what is observed in the folding-only model (Model 1), when two destabilizing mutations are combined in the same protein, the change in concentration of the protein bound to the ligand is often larger than the additive or log-additive expectation i.e., there is negative epistasis” (Figure 3c and Supplementary Figure 3a, b).” However, it is not clear how does the figure show ‘often larger’. Is it because the area shaded by dark green encompasses more in the observed vs expected plot because the phenotype is more severe? This should also be clearly stated in the main text. This is true for all the figures presented in the manuscript.

Authors should explain in the main text why is the expectation the same for the folding and ligand binding mutants in Fig 3?

In general results are very poorly explained. The authors should explain what each panel shows and how it supports their conclusions. For example, there is one sentence line 235-236 that is used to state “When mutations are combined in the same allele in Model 2, the outcome differs

depending on whether the mutations affect folding or binding (Figure 3c, e, f).” There is not enough information provided here. The authors should go through each subpanel and explain this statement. This is true for all the results presented in this manuscript.

The entire manuscript relies on simulations but it would be good to test some or all of these simulations using real biological data. This will strongly enhance the manuscript.

Reviewer #2 (Remarks to the Author):

This is an interesting piece that highlights the complexity of the genetic phenomena that can be produced by even the simplest biophysical models. In particular, the authors investigate a well-known and influential model of protein folding from the perspective of diploid genetics, and investigate how adding even a small degree of complexity (binding of a ligand, a nonlinear relationship between protein levels and phenotype) can dramatically alter the possible genetic patterns.

From a purely scientific perspective, these results are fascinating, and I feel they push the field forward in terms of calibrating our null expectations for how mutations interact. However, there are serious discrepancies between the way that epistasis and dominance are being discussed here and the standard meaning of these terms as applied to the analysis of a biallelic diploid system. Specifically, the authors focus here on the comparison between the effects of two single heterozygous mutations and a double heterozygote, where the double heterozygote either consists of both mutants on the same chromosome (whose phenotype they use to compute epistasis) or the two mutants on different chromosomes (which they use to compute dominance). In contrast, the standard view would consider epistasis and dominance to be characteristics of the phenotypes of 9 genotypes (the 4 double homozygotes, 4 single heterozygotes, and 1 double heterozygote), as described e.g. by Mackay 2014 or Sackton and Hartl 2016 (and typically also distinguish between the various forms of epistasis such as additive by additive, dominance by additive, etc.). The distinction between the two types of double heterozygotes focused on here would then be addressed under the topic of the influence of gametic phase disequilibrium (see e.g. Lynch and Walsh Chapter 5). More generally, the analysis of epistasis here where the interaction is evaluated in a heterozygous state with the wildtype allele strongly distorts the nature of the epistatic interaction when the wild-type is dominant. This leads to many of the Figure panels being bizarrely uninformative e.g. the almost solid green panels in Figure 3c and Figure 4c.

The authors should address these difficulties by reconsidering which properties of their models correspond to epistasis and dominance as these terms have been defined in the existing literature on diploid models and provide the reader with guidance on the broader issue of how these terms have been and ought to be defined.

Other comments:

1. I could not find a description of the meaning of the multiple lines and shading of Figure 2I and similar.
2. The ligand binding model described here was previously investigated by Manhart and Morozov PNAS 2015 with a different emphasis.
3. The abstract should be more explicit that the paper consider two and three state thermodynamic models of protein folding and ligand binding.
4. The degree of rigor should be increased in certain passages to clarify whether the claim is what is observed for a certain set of parameters values or a necessary consequence of some more generic situation (e.g. lines 313-314 "A concave function shifts within-allele interactions to less negative or less positive values").

RESPONSE TO REVIEWER COMMENTS

Reviewer #1 (Remarks to the Author):

Thank you for the opportunity to review “Dominance vs. epistasis: the biophysical origins and plasticity of genetic interactions within and between alleles” by Xie et al. This manuscript discusses how to predict phenotypes as a result of multiple mutations within and between alleles that lead to epistasis vs dominance. Authors use simulations of a simple biophysical system to examine mutations within and between alleles that affect folding or ligand binding. They show that the emergence of epistasis or dominance depends on the biophysical property of the mutation highlighting the difficulty in predicting phenotypes.

This study investigates how mutations combine to impact phenotypes and result in epistatic vs dominant effects. This addresses an important question in the field of genetic interactions and provides novel insight about how mutations that affect different biochemical properties combine to affect phenotypes. This is an interesting study although the manuscript is very difficult to read and does not provide a sufficient level of detail necessary to understand the results.

We thank Reviewer #1 for the enthusiasm and for recognizing the novelty and importance of our study. We have endeavored to make the revised manuscript easier to read. Please see the point-to-point responses to the specific comments.

Specific comments:

Figure legends should explain what the different colours, outlines within the heatmaps and dashed lines mean for every panel. Currently, it is not clear what the figures show. For example, authors indicate that on lines 208-211 “Similar to what is observed in the folding-only model (Model 1), when two destabilizing mutations are combined in the same protein, the change in concentration of the protein bound to the ligand is often larger than the additive or log-additive expectation i.e., there is negative epistasis” (Figure 3c and Supplementary Figure 3a, b).” However, it is not clear how does the figure show ‘often larger’. Is it because the area shaded by dark green encompasses more in the observed vs expected plot because the phenotype is more severe? This should also be clearly stated in the main text. This is true for all the figures presented in the manuscript.

We thank the reviewer for pointing this out. We added more information to the figure legends including what the colors and dashed lines indicate. In addition, we added more information throughout the main text to make the messages clearer including at the indicated lines.

“Similar to what is observed in the folding-only model (Model 1), when two destabilizing mutations are combined in the same protein, the decrease in concentration of the protein bound to the ligand is often larger than the additive or log-additive expectation i.e., there is negative epistasis (blue shaded areas in Figure 3c and Supplementary Figure 3a, b).”

Authors should explain in the main text why is the expectation the same for the folding and ligand binding mutants in Fig 3?

The expectations are the same for folding and binding mutants because they are calculated from the phenotypic values of the mutations: mutations with the same phenotypic effects have the same additive or log-additive expectation irrespective of the underlying molecular mechanism. We have added some text on page 9 to repeat this point:

“In Figure 3c and 3d, we compare the observed and expected double mutant phenotypes when combining mutations affecting either folding or binding (the expected values are the same for folding and binding mutations because the additive and log-additive expectations are calculated from the phenotypic values).”

In general results are very poorly explained. The authors should explain what each panel shows and how it supports their conclusions. For example, there is one sentence line 235-236 that is used to state “When mutations are combined in the same allele in Model 2, the outcome differs depending on whether the mutations affect folding or binding (Figure 3c, e, f).” There is not enough information provided here. The authors should go through each subpanel and explain this statement. This is true for all the results presented in this manuscript.

We added more information and rewrote this part as follows:

“When mutations are combined in the same allele in Model 2, the outcome differs depending on whether the mutations affect folding or binding. That is, combining single mutants with the same phenotypic value but different underlying causal biophysical mechanisms results in a different outcome in a double mutant (compare “Folding mutants” and “Binding mutants” columns in Figure 3c). As shown in Figure 3e, folding mutant phenotypes deviate from the expected additive phenotypes more than binding mutant phenotypes, resulting in stronger negative interaction scores (Figure 3f).

We have also modified the legends to Figure 3c, d and Figure 4.

The entire manuscript relies on simulations but it would be good to test some or all of these simulations using real biological data. This will strongly enhance the manuscript.

In the revised manuscript we have included experimental data where we quantified the interactions within and between alleles in a model system. These results are now included in Figure 2 and described in the results and methods sections:

“To evaluate these results experimentally, we expressed two copies of the N-terminal domain of the bacteriophage lambda repressor CI fused to GFP (Figure 2l, m, Supplementary Figure 7). This protein domain does not dimerize. We selected eight individual mutants of CI from our earlier study (Li et al., 2019) (see the Methods section for more details) and quantified the total protein fluorescence when the mutations are combined in two different copies or in the same copy of the gene (Figure 2n, o, Supplementary Figure 8). In agreement with the expectation from

our simulations, mutations typically combined additively between alleles but have lower than expected concentration when combined within the same allele (Figure 2p, q).”

We also now mention in the discussion that examples of variants reported as both dominant and recessive in different individuals exist in the human clinical genetics literature:

“As an example, our results suggest that it is not unreasonable to expect disease-causing alleles to switch from dominant to recessive depending upon the conditions, examined phenotype or individual. Indeed, examples exist in human clinical genetics of variants reported to be both dominant and recessive, for example in Marfan syndrome (Overwater et al., 2019).

Reviewer #2 (Remarks to the Author):

This is an interesting piece that highlights the complexity of the genetic phenomena that can be produced by even the simplest biophysical models. In particular, the authors investigate a well-known and influential model of protein folding from the perspective of diploid genetics, and investigate how adding even a small degree of complexity (binding of a ligand, a nonlinear relationship between protein levels and phenotype) can dramatically alter the possible genetic patterns.

From a purely scientific perspective, these results are fascinating, and I feel they push the field forward in terms of calibrating our null expectations for how mutations interact.

We thank the reviewer for their enthusiasm and positive comments about our work.

However, there are serious discrepancies between the way that epistasis and dominance are being discussed here and the standard meaning of these terms as applied to the analysis of a biallelic diploid system. Specifically, the authors focus here on the comparison between the effects of two single heterozygous mutations and a double heterozygote, where the double heterozygote either consists of both mutants on the same chromosome (whose phenotype they use to compute epistasis) or the two mutants on different chromosomes (which they use to compute dominance). In contrast, the standard view would consider epistasis and dominance to be characteristics of the phenotypes of 9 genotypes (the 4 double homozygotes, 4 single heterozygotes, and 1 double heterozygote), as described e.g. by Mackay 2014 or Sackton and Hartl 2016 and typically also distinguish between the various forms of epistasis such as additive by additive, dominance by additive, etc.). The distinction between the two types of double heterozygotes focused on here would then be addressed under the topic of the influence of gametic phase disequilibrium (see e.g. Lynch and Walsh Chapter 5).

After carefully reading the mentioned references, we realize that we did not convey clearly that the epistasis we are quantifying in this manuscript is intra-genic (within the same gene, different mutations) epistasis (Starr and Thornton, 2016, Protein Science; Domingo et al., 2019 Annual Review of Genomics and Human Genetics) rather than epistasis between variants in different genes (as in Mackay 2014 and Sackton and Hartl 2016).

Although our study is on a diploid system, it involves one gene locus with two mutations. In other words, it is a diploid system with one gene, four alleles (1-wildtype allele; 2-mutant A allele; 3-mutant B allele; 4-Mutant AB allele; this is shown in Figure 1). We have added text to the first section of the results section to clarify this point:

“Our study is of a monogenic diploid model system where epistasis refers to intragenic epistasis ((Bank et al., 2015, 2016; Domingo et al., 2019; Gonzalez et al., 2019; Starr & Thornton, 2016)). This should not be confused with the digenic diploid systems where epistasis between genes (intergenic epistasis) and dominance within the same gene are often studied (Mackay, 2013; Sackton & Hartl, 2016)”

More generally, the analysis of epistasis here where the interaction is evaluated in a heterozygous state with the wildtype allele strongly distorts the nature of the epistatic interaction when the wild-type is dominant. This leads to many of the Figure panels being bizarrely uninformative e.g. the almost solid green panels in Figure 3c and Figure 4c. The authors should address these difficulties by reconsidering which properties of their models correspond to epistasis and dominance as these terms have been defined in the existing literature on diploid models and provide the reader with guidance on the broader issue of how these terms have been and ought to be defined.

We introduce how we quantify interactions within an allele (epistasis) and between 2 alleles (dominance) at the start of the results section on page 5 (copied below). The goal of this study is to compare how mutations interact within and between alleles. Using the same interaction metrics for within and between allele interactions makes this comparison much simpler and more transparent. Measures of epistasis typically use a log-additive null model whereas measures of dominance typically use an additive model. Throughout the manuscript we therefore quantify and present interactions using both of these null models, allowing direct comparison of the interaction strengths within and between alleles.

The mostly solid green panels in Figures 3c and 4c are caused by the strong negative epistasis between mutations affecting folding or binding (there is a sigmoidal relationship between binding and free energy that is similar to the relationship between folding and free energy shown in Figure 2C: two mutations that cause a small increase in energy therefore combine to cause a larger than additive or log-additive change in binding). We think it is important to show this and to contrast it to the between allele interactions of the same mutations in Figure 3d and 4c.

“Quantifying how variants interact when they are combined requires the specification of a null model for independent effects. The interactions between variants within the same allele of a gene are typically referred to as epistasis or genetic interactions with a log-additive (or sometimes additive) null model used as the expected outcome (Figure 1b) (Domingo et al., 2019; Mani et al., 2008). In contrast, interactions between variants in different alleles are typically quantified as dominance, which uses an additive expected outcome (Figure 1c, Supplementary Figure 1) (Falconer & Falconer, 1989; Wright, 1934). Therefore, when comparing how mutations interact within and between alleles of a gene, throughout this manuscript we quantify interactions using both of these null models – additive and log-additive – as well as directly compare the phenotypes of double mutants (Figure 1d, e). Our study is of a monogenic diploid model system

where epistasis refers to intragenic epistasis ((Bank et al., 2015, 2016; Domingo et al., 2019; Gonzalez et al., 2019; Starr & Thornton, 2016)). This should not be confused with the digenic diploid systems where epistasis between genes (intergenic epistasis) and dominance within the same gene are often studied (Mackay, 2013; Sackton & Hartl, 2016).”

Other comments:

1. I could not find a description of the meaning of the multiple lines and shading of Figure 2I and similar.

We have added this information to the legend of Figure 2 and similar figures.

2. The ligand binding model described here was previously investigated by Manhart and Morozov PNAS 2015 with a different emphasis.

We thank the reviewer for the information. We added citation to the paper in the revised manuscript.

3. The abstract should be more explicit that the paper consider two and three state thermodynamic models of protein folding and ligand binding.

We have added the following text: “Here based on analyses of thermodynamic models of protein folding and ligand-binding, we show that even in very simple biophysical systems, interactions between mutations are frequent, context-dependent and different when variants are combined within and between alleles.”

4. The degree of rigor should be increased in certain passages to clarify whether the claim is what is observed for a certain set of parameters values or a necessary consequence of some more generic situation (e.g. lines 313-314 "A concave function shifts within-allele interactions to less negative or less positive values").

We examined the robustness of our claim by changing parameter values in all the examined nonlinear functions (the concave, convex and sigmoidal functions). As shown in the revised Supplementary Figure 6 and highlighted at the end of the results section, our conclusions are robust to parameter values.

“To examine how dependent our conclusions are on the parameter values defining the non-linear curves, we repeated our analysis with five different curvatures (slopes) for each of the concave, convex and sigmoidal curves (Supplementary Figure 6a). We observe stronger between-allele negative interactions with steeper slopes of concave or sigmoidal curves (Supplementary Figure 6b, c) but the direction of the changes is independent of the parameter values (Supplementary Figure 6b, c).”

REVIEWER COMMENTS

Reviewer #1 (Remarks to the Author):

Authors thoroughly addressed all the comments.

Reviewer #2 (Remarks to the Author):

The revisions to this manuscript do not address the major issue with this manuscript from the first round of review, which is that the terms “dominance” and “epistasis” in the title, abstract, and throughout the text are used incorrectly. This is deeply misleading and the manuscript is simply not publishable in its current form. The authors need to review existing genetic terminology to provide accurate descriptions of the results presented here. For example, it would be accurate to say that the authors consider complementation phenotypes for mutations within a single protein coding gene, where specifically the phenotypes of the cis- and trans- double heterozygotes are compared to the expected phenotype based on the effects of the two mutations as observed in the single heterozygotes. There are many other accurate ways to describe the results, but especially the term “dominance” as used here is almost completely unrelated to its standard definition.

The authors attempted to address this issue by inserting a brief passage stating “Our study is on a monogenic diploid model system where epistasis refers to intragenic epistasis. This should not be confused with the eugenic diploid systems where epistasis between genes (intergenic epistasis) and dominance within the same gene are often studied.” However, their treatment is not correct even as an analysis of a single locus diploid system. How to convert between a 2 locus di-allelic and 1 locus multi-allelic description is well known (see e.g. Shaid 2003 Evaluating associations of haplotypes with traits), and the dominance of the double mutant haplotype is correctly determined based on where the AB/ab heterozygote falls relative to the AB/AB and ab/ab homozygotes, rather than based on the complex 4 genotype comparison considered here. Indeed the 4 genotype comparison hardly has any relation to dominance as commonly understood, since it is defined as e.g. $Ab/aB - AB/aB - AB/Ab + AB/AB$ (Figure 1e of the current manuscript), which only considers the phenotype of one homozygote!

In fact, the analysis of how two heterozygous mutations combine has long been studied in genetics under the topic of “complementation”. Although perhaps omitted in a first course in genetics, this is standard material in an advanced course in classical genetics and is reviewed in e.g. Chapter 11 of the CRC reference volume “Advanced Genetics” or the Hawley and Gilliland 2006 review in Genetics. In this literature, the distinction between the two doubly homozygous configurations is a key object of study, and both are constructed in the so-called cis-trans complementation test, where the cis configuration has both mutations on the same chromosome (the genotype relevant to “epistasis” in the current

manuscript) and the trans configuration has the two mutations on different chromosomes (the genotype relevant to “dominance” in the current manuscript). Indeed, the analysis of the relations between more than 2 mutations in a single gene has historically been common and discussed as a “complementation map”, which provided early insight into the functioning of proteins with multiple domains and the construction of multimeric assemblies (see e.g. the CRC book chapter), and the main purpose of the complementation test is specifically to determine which mutations lie in the same gene.

A different point of view on these issues that is perhaps more common in the contemporary literature is to refer to the difference between the cis and trans configurations of the double heterozygotes as a “phase effect”. Such effects have been of great interest, particularly in human genetics over the last 10 years, see e.g. Towhey et al 2011 Nature Reviews Genetics. The recognition of these phase effects has led to considerable interest in designing association studies that take these effects into account, such as gene based association studies and haplotype based association studies (e.g. Cox et al. 2013 Genetics Research or Sanjek, Long and Thornton 2017 PLOS Genetics). This literature provides a different way of describing and contextualizing the relevance of the biophysical arguments presented in the current manuscript.

The authors should correct their use of genetic terminology in describing their results. Since these issues will likely be unfamiliar to non-expert readers, the authors should also provide the reader with accurate guidance about the relationship of these results to the various concepts described above, including (1) the standard treatment of 2 locus diallelic models, (2) the cis-trans complementation test and complementation maps, and (3) phase effects and their current treatment in human genetics.

REVIEWER COMMENTS

Reviewer #1 (Remarks to the Author):

Authors thoroughly addressed all the comments.

Thank you.

Reviewer #2 (Remarks to the Author):

The revisions to this manuscript do not address the major issue with this manuscript from the first round of review, which is that the terms “dominance” and “epistasis” in the title, abstract, and throughout the text are used incorrectly. This is deeply misleading and the manuscript is simply not publishable in its current form. The authors need to review existing genetic terminology to provide accurate descriptions of the results presented here. For example, it would be accurate to say that the authors consider complementation phenotypes for mutations within a single protein coding gene, where specifically the phenotypes of the cis- and trans- double heterozygotes are compared to the expected phenotype based on the effects of the two mutations as observed in the single heterozygotes. There are many other accurate ways to describe the results, but especially the term “dominance” as used here is almost completely unrelated to its standard definition.

The authors attempted to address this issue by inserting a brief passage stating “Our study is on a monogenic diploid model system where epistasis refers to intragenic epistasis. This should not be confused with the eugenic diploid systems where epistasis between genes (intergenic epistasis) and dominance within the same gene are often studied.” However, their treatment is not correct even as an analysis of a single locus diploid system. How to convert between a 2 locus di-allelic and 1 locus multi-allelic description is well known (see e.g. Shaid 2003 Evaluating associations of haplotypes with traits), and the dominance of the double mutant haplotype is correctly determined based on where the AB/ab heterozygote falls relative to the AB/AB and ab/ab homozygotes, rather than based on the complex 4 genotype comparison considered here. Indeed the 4 genotype comparison hardly has any relation to dominance as commonly understood, since it is defined as e.g. $Ab/aB - AB/aB - AB/Ab + AB/AB$ (Figure 1e of the current manuscript), which only considers the phenotype of one homozygote!

In fact, the analysis of how two heterozygous mutations combine has long been studied in genetics under the topic of “complementation”. Although perhaps

omitted in a first course in genetics, this is standard material in an advanced course in classical genetics and is reviewed in e.g. Chapter 11 of the CRC reference volume “Advanced Genetics” or the Hawley and Gilliland 2006 review in Genetics (<https://academic.oup.com/genetics/article/174/1/5/6061035>).

In this literature, the distinction between the two doubly homozygous configurations is a key object of study, and both are constructed in the so-called cis-trans complementation test, where the cis configuration has both mutations on the same chromosome (the genotype relevant to “epistasis” in the current manuscript) and the trans configuration has the two mutations on different chromosomes (the genotype relevant to “dominance” in the current manuscript). Indeed, the analysis of the relations between more than 2 mutations in a single gene has historically been common and discussed as a “complementation map”, which provided early insight into the functioning of proteins with multiple domains and the construction of multimeric assemblies (see e.g. the CRC book chapter), and the main purpose of the complementation test is specifically to determine which mutations lie in the same gene.

A different point of view on these issues that is perhaps more common in the contemporary literature is to refer to the difference between the cis and trans configurations of the double heterozygotes as a “phase effect”. Such effects have been of great interest, particularly in human genetics over the last 10 years, see e.g. Towhey et al 2011 Nature Reviews Genetics. The recognition of these phase effects has led to considerable interest in designing association studies that take these effects into account, such as gene based association studies and haplotype based association studies (e.g. Cox et al. 2013 Genetics Research or Sanjek, Long and Thornton 2017 PLOS Genetics) . This literature provides a different way of describing and contextualizing the relevance of the biophysical arguments presented in the current manuscript.

The authors should correct their use of genetic terminology in describing their results. Since these issues will likely be unfamiliar to non-expert readers, the authors should also provide the reader with accurate guidance about the relationship of these results to the various concepts described above, including (1) the standard treatment of 2 locus diallelic models, (2) the cis-trans complementation test and complementation maps, and (3) phase effects and their current treatment in human genetics.

We apologise for the confusion and for misunderstanding your previous comments.

In the revised manuscript we now also quantify all of the between allele interactions as dominance using the dominance index. We agree this provides needed consistency

with the literature. We also do not refer to any other interaction metrics as dominance and we have included equations throughout the text defining our interaction metrics to avoid any ambiguity.

However, the core goal of this study is to compare how the same variants interact within one allele of a gene compared to when each is present in a different allele. Such a comparison can only be sensibly made if the same metric is used to quantify the within-allele interactions and the between-allele interactions. Otherwise one would be comparing apples with oranges and many differences will simply be due to the different metrics being used to quantify the interactions in the within- vs. between-allele arrangements. Throughout the manuscript we therefore deliberately use the same two simple measures of genetic interaction, which are identical to the additive and log-additive metrics of epistasis that are widely used by many different communities.

We hope that this is now clarified in the wording of the revised manuscript.

REVIEWERS' COMMENTS

Reviewer #2 (Remarks to the Author):

The authors have satisfactorily addressed my comments from the previous round of review. I have a few more minor comments below that would improve the clarity of the manuscript.

Minor comments:

Line 90. Citation is not clear here. Wright and Kacser & Burns use h whereas Omholt (and Falconer?) use the degree of dominance. Better to explain that there are two conventions but that they are equivalent.

Line 189. There are issues with both the presentation and biological assumptions here. The condition that the AB mutant has an expected phenotype greater than $1/2$ would seem to preclude dominant mutations entirely. This might be reasonable for the biophysical models considered here but not for general definitions.

Also, the inequalities in equations 7 and 8 do not correspond to the intended model, since the systems of inequalities are inconsistent for strong single mutant effects. Rather the idea is that e.g. $W_{\text{exp}} = \text{Max}(1/2, W_{\text{single1}} + W_{\text{single2}} - W_{\text{wt}})$. I suggest rewriting to clarify.

Line 528. Typo "as shown in the following equations Eq. (6)."

Line 553. The notation throughout this section for the concentration of gene products is confusing. Are α_1 and α_2 the maternal and paternal alleles? Or does one or the other concentration double in the case of a homozygote? Can $\alpha_i = \text{WT}$? Is $[X_{\text{F},\alpha_i}] = [X_{\text{F},\text{WT}}]$ if $\alpha_i = \text{WT}$? In that case $[X_{\text{F},\text{WT}}]$ is only half the concentration of the folded gene product in a wild-type genotype, and equation 13 is missing a factor of 2 in the denominator (this factor appears to be present in equation 14). There are similar issues for model 2 and e.g. Equation 23.

Line 578 Typo, α^{WT} should be α^{B} here.

Line 951 "ully"

Line 999 AU is usually “arbitrary units” not “artificial units”

RESPONSE TO REVIEWERS' COMMENTS

Reviewer #2 (Remarks to the Author):

The authors have satisfactorily addressed my comments from the previous round of review. I have a few more minor comments below that would improve the clarity of the manuscript.

Thanks for the thorough revision. We made all the requested modifications accordingly.

Minor comments:

Line 90. Citation is not clear here. Wright and Kacser & Burns use h whereas Omholt (and Falconer?) use the degree of dominance. Better to explain that there are two conventions but that they are equivalent.

In the revised manuscript,

“...with any deviance from this additive expectation quantified as the dominance index^{6,7} or degree of dominance (Eq.6)^{8,9}. The former uses one allele as the reference and scores above or below 0.5 indicate the reference allele is recessive or dominant, respectively. On the other hand, the degree of dominance does not set a reference allele and 0 indicates no dominance, a positive value indicates that the allele with better function is dominant, and vice versa. Complete dominance or recessivity (where the heterozygote phenotype is the same as either parent) leads to an absolute value of 1.”

Line 189. There are issues with both the presentation and biological assumptions here. The condition that the AB mutant has an expected phenotype greater than 1/2 would seem to preclude dominant mutations entirely. This might be reasonable for the biophysical models considered here but not for general definitions.

Also, the inequalities in equations 7 and 8 do not correspond to the intended model, since the systems of inequalities are inconsistent for strong single mutant effects. Rather the idea is that e.g. $W_{exp} = \text{Max}(1/2, W_{single1} + W_{single2} - W_{wt})$. I suggest rewriting to clarify.

We rewrote the equation and modified the paragraph accordingly:

$$\log(W_{exp_log}) = \text{Max} \left(\log(C), \log(W_{\alpha^A/\alpha^{WT}}) + \log(W_{\alpha^B/\alpha^{WT}}) - \log(W_{\alpha^{WT}/\alpha^{WT}}) \right) \quad (7)$$

$$W_{exp_add} = \text{Max}(C, (W_{\alpha^A/\alpha^{WT}} + W_{\alpha^B/\alpha^{WT}} - W_{\alpha^{WT}/\alpha^{WT}})) \quad (8)$$

$$C = \begin{cases} 0.5 \text{ for } \alpha^{AB}/\alpha^{WT} \\ 0 \text{ for } \alpha^A/\alpha^B \end{cases}$$

The wild-type phenotype $W_{\alpha^{WT}/\alpha^{WT}}$ is set to 1 and complete-loss of function as 0. For within-allele combinations of mutations (α^{AB}/α^{WT}), we set the lower bound of a phenotype to be half of the wild-type phenotype as the second allele of the gene is always functional (Figure 1b) (Eq. 7, 8), which is not a general definition but a reasonable treatment for our biophysical model.

Line 528. Typo “as shown in the following equations Eq. (6).”

Corrected to “as shown in Eq. (5)”

Line 553. The notation throughout this section for the concentration of gene products is confusing. Are alpha 1 and alpha 2 the maternal and paternal alleles? Or does one or the other concentration double in the case of a homozygote? Can $\alpha_i = WT$? Is $[X_F, \alpha_i] = [X_F, WT]$ if $\alpha_i = WT$? In that case $[X_F, WT]$ is only half the concentration of the folded gene product in a wild-type genotype, and equation 13 is missing a factor of 2 in the denominator (this factor appears to be present in equation 14). There are similar issues for model 2 and e.g. Equation 23.

We added the following information to clarify:

“In this model, the protein of interest (X) expressed from each allele (allele α_i , $i \in \{1, 2\}$ – one maternal and the other paternal copy respectively, and the alleles are allowed to be the wild type) ...”

We also added the missing factor of 2 in denominators of Eq. 13 and 23. Those were lost during the revision editing process. We carefully examined all the equations and made sure of no errors.

Line 578 Typo, α^{WT} should be α^B here.

Corrected to α^B .

Line 951 “ully”

Corrected to “Fully”

Line 999 AU is usually “arbitrary units” not “artificial units”

Corrected to “arbitrary units”.